# ARCHITECTURAL PLASTICITY FOR CONTINUAL LEARNING

## ABSTRACT

Neural networks for continual reinforcement learning (CRL) often suffer from plasticity loss, i.e., a progressive decline in their ability to learn new tasks arising from increased churn and Neural Tangent Kernel (NTK) rank collapse. We propose InterpLayers, a drop-in architectural solution that combines a fixed, parameter-free reference pathway with a learnable projection pathway using input-dependent interpolation weights. Without requiring algorithmic adaptation, InterpLayers conserve gradient diversity and constrain output variability by integrating stable and adaptive computations. We provide theoretical guarantees for bounded churn and show that, under mild assumptions, InterpLayers prevent NTK rank collapse through a non-zero rank contribution from the interpolation weights. Across environments with distributional shifts including permutation, windowing, and expansion, InterpLayer variants (convonly, fullinterp) consistently mitigate performance degradation compared to parameter-matched baselines. Furthermore, lightweight modifications such as dropout improve performance, especially under gradual shifts. These results position InterpLayers as a simple, complementary solution for maintaining plasticity in CRL.

## 1 INTRODUCTION

Continual reinforcement learning (CRL) requires agents to adapt to a non-stationary stream of tasks without external resets or explicit knowledge of task boundaries. Yet neural networks trained in this setting suffer from *plasticity loss*: their ability to adapt to new tasks diminishes over time. Plasticity loss has been attributed to several interacting factors, including rank collapse of the Neural Tangent Kernel (NTK) (Lyle et al., 2024), unbounded weight growth (Lyle et al., 2023), and representational drift or churn that destabilizes previously acquired knowledge (Tang et al., 2025).

Most existing solutions intervene at the algorithmic level. Reset-based strategies reinitialize parameters on a fixed schedule (Igl et al., 2020; Nikishin et al., 2022; 2023). Continuous plasticity methods modify the optimization process itself, e.g., shrink-perturb (Ash & Adams, 2020), ReDo (Sokar et al., 2023), or regenerative regularization (Kumar et al., 2023). Constraint-based approaches rely on normalization, clipping, or masking to restrict parameter dynamics (Ba et al., 2016; Abbas et al., 2023; Elsayed et al., 2024). While effective, these methods share limitations, including: (i) requiring task boundary information or chosen reset schedules; (ii) introducing hyperparameters such as reset frequencies, perturbation magnitudes, or regularization strengths; (iii) acting externally to the architecture, often outside the optimization framework.

Here, we offer a distinct alternative by addressing plasticity loss directly at the architectural level, without the need for interventions during training. Our method enhances standard network layers with additional pathways to build *Interpolation Layers* (InterpLayers). Each layer combines a fixed, parameter-free reference pathway that preserves stable representations throughout training and a learnable projection pathway that adapts through backpropagation, connected via input-dependent interpolation weights. By dynamically interpolating between these pathways, the network maintains representational stability while preserving the capacity for adaptive learning. Unlike ResNet-like skip connections, which only diversify gradient flow, or parameter-efficient tuning methods such as LoRA, which fine-tune computational efficiency, InterpLayers create a self-regulating mechanism that balances stability and plasticity without external intervention. Moreover, compared to algorithmic approaches like soft-shrink-perturb with layer normalization (Juliani & Ash, 2024), Inter-

pLayers require minimal computational overhead and no additional schedules or hyperparameters. Designed as orthogonal components to current solutions for plasticity loss, they can be integrated seamlessly into existing architectures or combined with intervention mechanisms.

We evaluate InterpLayers both theoretically and empirically. We perform a theoretical analysis to investigate how InterpLayers impact churn and NTK rank, demonstrating that these properties are enhanced by the interpolation mechanism between reference and projection pathways. For empirical evaluation, we evaluate the performance of InterpLayers over standard baselines for ProcGen tasks as described in Juliani & Ash (2024). We also investigate the performance of InterpLayers when combined with dropout (Srivastava et al., 2014) and discuss how to effectively combine InterpLayers orthogonally with other methods that tackle plasticity loss. We show that InterpLayers are effective in preventing plasticity loss and can be a direction for future architectural solutions for continual learning.

Our main contributions can be denoted as follows.

1. We introduce InterpLayers as drop-in replacements for conventional neural network layers. InterpLayers splits the layer input into a reference and a projection pathway that are further interpolated to obtain the layer's output.

2. We show that InterpLayers bound representational drift through controlled interpolation, limit churn growth via pathway stability, and maintain NTK rank under specific assumptions. These guarantees emerge from architectural constraints rather than external interventions.

3. Across ProcGen distribution shifts spanning pixel permutations, level expansion, and sequential task changes, InterpLayers preserve performance where standard multi-layer perceptron (MLP) layers collapse. We also empirically compare the performance of InterpLayers with other interventions to counter plasticity loss.

## 2 RELATED WORKS

### 2.1 ALGORITHMIC APPROACHES TO MITIGATE PLASTICITY LOSS

**Reset-based interventions.** Periodic parameter reinitialization has often been applied to counter plasticity loss. Igl et al. (2020) proposed resetting only the final layer to preserve learned features while restoring adaptability. Nikishin et al. (2022) showed that resetting selected network parameters on a fixed schedule restores the network's capacity to learn. Later, Nikishin et al. (2023) has shown that resetting the entire network leads to maintenance of plasticity at the cost of losing prior knowledge. To implement these methods, reset schedules and selecting which parameters to reinitialize is needed.

**Continuous plasticity upkeep.** Other methods continuously regulate plasticity during training. Sokar et al. (2023) proposed ReDo, which periodically resets inactive neurons. A continual backpropagation method was presented by Dohare et al. (2024), which adds a step to backpropagation where a small fraction of neurons are continuously reinitialized based on a utility metric. Ash & Adams (2020) applied a shrink-and-perturb methodology to the network after each update to scale down the weights and add noise in order to maintain plasticity. To prevent unbounded weight drift, Kumar et al. (2023) used regenerative regularization applying L2 penalties to weights. Abbas et al. (2023) showed that increasingly sparse activation patterns diminish gradients, causing plasticity loss. To prevent this, they introduced CReLU as a modified activation function to mitigate sparsity.

**Normalization and constraint-based methods.** Another approaches alleviate plasticity loss by constraining the network dynamics. Lyle et al. (2023) showed that LayerNorm can slow down plasticity loss, as it helps to maintain NTK rank. Elsayed et al. (2024) investigated weight clipping to provide an upper bound to parameter growth. To stabilize optimization, Miyato et al. (2018) has shown that spectral normalization can constrain Lipschitz constants. Even though plasticity loss is reduced, representational capacity is also affected by the constraints added by these methods. Recently, Lee et al. (2025) proposed an architecture named SimbaV2, which constrains weight growth and feature norm by hyperspherical normalization and makes use of reward scaling to maintain gradient stability. Additionally, Nauman et al. (2024) introduced the BRO algorithm, which combines

LayerNorm, weight decay, and full-parameter resets, to scale the vanilla SAC model to seven times its size, improving performance while maintaining sample efficiency.

## 2.2 ARCHITECTURAL MECHANISMS FOR STABILITY IN NEURAL NETWORKS

Various innovations in neural network architectures have been proposed to balance stability and plasticity, even though they have not been directly applied to continual learning. Skip connections and residual pathways have been vastly investigated to create gradient highways and regulate the information flows in computer vision (He et al., 2016; Srivastava et al., 2015). Gating mechanisms for controlling information flow have also been highly effective in natural language processing architectures (Hochreiter & Schmidhuber, 1997; Cho et al., 2014). Networks that generate specific parameters conditioned on input features, such as HyperNetworks (Ha et al., 2016), have also been investigated to introduce architectural flexibility in meta-learning tasks. Here, these methods serve as a foundation for the theoretical modeling of InterpLayers, which introduce an asymmetry by keeping one pathway fixed and parameter-free, thereby achieving input specificity and representational stability.

## 2.3 THEORETICAL UNDERSTANDING OF PLASTICITY LOSS

Recent works have also explored key theoretical features to enhance understanding of plasticity loss in neural networks. Lyle et al. (2024) showed that the effective NTK rank is strongly linked with the ability of the network to adapt in a continual learning setting. Specifically, they demonstrate that NTK rank collapse correlates with a decrease in performance. The unconstrained drift of internal network representation has also been described as a cause for catastrophic forgetting in CRL by Kumar et al. (2023). The instability of network outputs, i.e., *churn*, is investigated by Tang & Berseth (2024); Tang et al. (2025) as an important factor in plasticity loss. In addition to these metrics, Lewandowski et al. (2023) showed that a decrease in curvature directions is another indicator of plasticity loss in neural networks. Based on these findings, we theoretically investigate the effects of InterpLayers on churn and effective NTK rank.

## 3 METHODS

### 3.1 PRELIMINARIES

We consider an agent that learns in a CRL environment interacting with a sequence of tasks $\{\mathcal{M}_1, \mathcal{M}_2, ..., \mathcal{M}_K\}$ following a Markov Decision Process (MDP), where each $\mathcal{M}_i = (\mathcal{S}_i, \mathcal{A}_i, P_i, r_i, \gamma)$ may have different state spaces $\mathcal{S}_i$, action spaces $\mathcal{A}_i$, transition dynamics $P_i$, and reward functions $r_i$. The tasks are separated by distribution shifts, which can range from small changes, e.g., reinitializing the environment with a new random seed, to substantial changes, e.g., permutations on the observation axis that completely modify the input distribution. At each timestep $t$, the agent observes state $s_t$, selects action $a_t$ according to policy $\pi_\theta(a|s)$, receives reward $r_t$, and transitions to state $s_{t+1}$. The policy $\pi_\theta(a|s)$ is parameterized by a neural network with weight parameters $\theta$ and trained via backpropagation.

In a continual learning setting, the current task $\mathcal{M}$ is changed after a fixed number of environment steps. The agent is given no information about task boundaries or identities, so it does not know which task it has to solve at a given moment. The agent should adapt to a new task by modifying its set of parameters $\theta$ online, having a shared policy for multiple tasks. The policy does not store past experiences in another data structure to sample from during training. In this way, the policy should maintain a balance between stability (preserving knowledge) and plasticity (acquiring new knowledge) in a non-stationary environment.

### 3.2 THE INTERPOLATION LAYER

As an architectural solution to tackle plasticity loss, we introduce InterpLayers (Figure 1), which are task-agnostic, require no additional hyperparameters, and can be seamlessly integrated into existing neural network architectures.

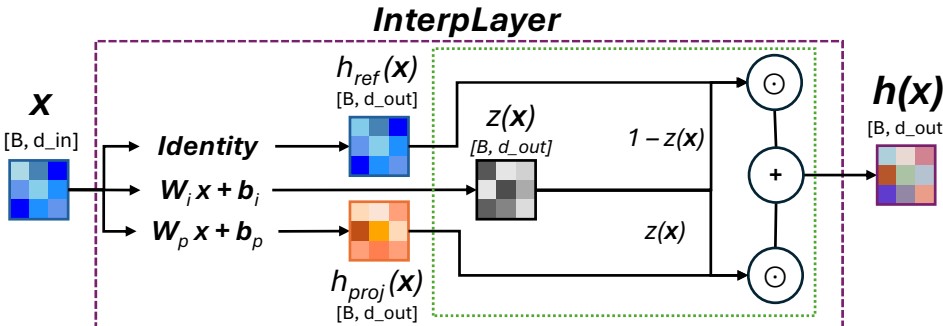

Figure 1: **The InterpLayer Architecture.** The input $\mathbf{x}$ is processed through a fixed reference pathway $h_{\mathrm{ref}}(\mathbf{x})$ and a learnable projection pathway $h_{\mathrm{proj}}(\mathbf{x})$. The learnable interpolation weights $z(\mathbf{x})$ dynamically interpolate the outputs from both pathways to produce the output $\mathbf{h}(\mathbf{x})$.

**Core mechanism.** Each InterpLayer splits computation into two complementary pathways: (i) a *reference pathway* given by a fixed, parameter-free mapping (identity, sparse selection, or padding when dimensions differ); and (ii) a *projection pathway* with standard learnable parameters. Learnable interpolation weights then combine both outputs, allowing the network to learn when to rely on preservation and when to adapt. Mathematically, given an input $\mathbf{x} \in \mathbb{R}^d$, the InterpLayer output is given as

$$\mathbf{h}(\mathbf{x}) = (1 - z(\mathbf{x})) \odot h_{\mathrm{ref}}(\mathbf{x}) + z(\mathbf{x}) \odot h_{\mathrm{proj}}(\mathbf{x}), \tag{1}$$

where $\odot$ denotes element-wise multiplication and $h_{\mathrm{ref}}$, $h_{\mathrm{proj}}$, and $z(\mathbf{x})$ are defined as

$$h_{\mathrm{ref}}(\mathbf{x}) = \mathbf{P}\mathbf{x}, \quad (\mathbf{P} = \mathbf{I} \text{ when } d_{\mathrm{in}} = d_{\mathrm{out}}), \tag{2}$$

$$h_{\mathrm{proj}}(\mathbf{x}) = \phi(\mathbf{W}_p \mathbf{x} + \mathbf{b}_p), \quad \mathbf{W}_p, \mathbf{b}_p \text{ (learnable)}, \tag{3}$$

$$z(\mathbf{x}) = \sigma(\mathbf{W}_i \mathbf{x} + \mathbf{b}_i), \quad \mathbf{W}_i, \mathbf{b}_i \text{ (learnable)}, \tag{4}$$

$d_{\mathrm{in}}$ and $d_{\mathrm{out}}$ denote the input and output dimensionalities of the layer, $\phi$ is a non-linear activation function and $\sigma$ is a sigmoid layer.

**Definition of the individual structures.** The reference pathway functions as a deterministic, parameter-free module $P$ that preserves the geometric structure of the input. For linear layers, we implement $P$ using an *IdentityProject* block: if $d_{\mathrm{in}} = d_{\mathrm{out}}$, $P$ is the identity; if $d_{\mathrm{out}} < d_{\mathrm{in}}$, $P$ is a fixed Johnson–Lindenstrauss (Dasgupta & Gupta, 2003) projection with orthonormal rows ($PP^{\top} = I_{d_{\mathrm{out}}}$) constructed with a seed fixed per layer; and if $d_{\mathrm{out}} > d_{\mathrm{in}}$, $P$ performs zero-padding to preserve the identity structure. For convolutional layers, we use an *IdentityDownsample* block: when the spatial resolution changes (i.e. stride > 1), we use average pooling; when channel counts differ we apply channel slicing (if $c_{out} < c_{in}$ or padding (if $c_{out} > c_{in}$). These modules have no learnable parameters, remain fixed during training, and serve only to preserve spatial structure to have a stable reference for interpolation. In contrast, the projection pathway enables adaptation through standard learning, similarly to an MLP layer. The interpolation weights $z(\mathbf{x}) \in (0, 1)^h$ regulate the contribution of reference and projection, providing the network with a dynamic preservation–adaptation tradeoff. This mechanism is similar to input gates in GRUs (Cho et al., 2014), but has a key difference: $h_{\mathrm{ref}}$ is a fixed skip from the current input rather than a recurrent hidden state from the past. Weight magnitudes closer to 0 are related to $\mathbf{h}(\mathbf{x})$ being mostly represented by the reference, while weight magnitudes closer to 1 are related to $\mathbf{h}(\mathbf{x})$ being mostly represented by the projection.

**Integration to convolutional layers.** InterpLayers can replace standard MLP layers following Eqs. (1)-(4). For convolutional layers processing image data $\mathbf{X} \in \mathbb{R}^{C_{\mathrm{in}} \times H \times W}$ as part of the state space, $h_{\mathrm{ref}}$, $h_{\mathrm{proj}}$, and $z(\mathbf{x})$ are defined as

$$h_{\mathrm{ref}}(\mathbf{X}) = \mathbf{P}_r * \mathbf{X} \tag{5}$$

$$h_{\mathrm{proj}}(\mathbf{X}) = \phi(\mathbf{W}_p * \mathbf{X} + \mathbf{b}_p), \tag{6}$$

$$z(\mathbf{X}) = \sigma(\mathbf{W}_i \cdot \beta(\mathbf{X}) + \mathbf{b}_i), \tag{7}$$

where $*$ denotes a convolution operation and $\beta$ is a global average pooling operation to produce channel-wise interpolation.

## 3.3 THEORETICAL PROPERTIES OF INTERPLAYERS

We analyze the mathematical properties of InterpLayers, focusing on two key properties: bounded representational drift and preservation of gradient diversity.

### 3.3.1 BOUNDED REPRESENTATIONAL DRIFT

The dual-pathway structure of InterpLayers ensures that changes in the output remain bounded under parameter updates. For an update $\Delta\theta = (\Delta\theta_p, \Delta\theta_z)$, the first-order output change is given as

$$\Delta h(\mathbf{x}) = z(\mathbf{x}) \odot \Delta h_{\text{proj}}(\mathbf{x}) + \Delta z(\mathbf{x}) \odot [h_{\text{proj}}(\mathbf{x}) - h_{\text{ref}}(\mathbf{x})]. \tag{8}$$

This decomposition shows that updates are constrained. The projection pathway update is modulated by the interpolation weights $z(\mathbf{x}) \in (0,1)^h$, while the interpolation update is bounded by the pathway difference.

**Theorem 1** (Bounded Output Variability). *If $h_{proj}$ is $L_p$-Lipschitz in its parameters $\theta_p$ and $z$ is $L_z$-Lipschitz in $\theta_z$, then*

$$\|\Delta h(\mathbf{x})\|_2 \leq \|z(\mathbf{x})\|_\infty L_p \|\Delta\theta_p\|_2 + L_z \|\Delta\theta_z\|_2 D(\mathbf{x}), \tag{9}$$

*where $D(\mathbf{x}) = \|h_{proj}(\mathbf{x}) - h_{ref}(\mathbf{x})\|_2$.*

The proof is deferred to Appendix A.1. This bound implies that churn is polynomially bounded in training steps, in contrast to standard MLP layers where churn may grow unboundedly with parameter norms. This theorem makes use of the fact that the reference pathway is parameter-free at the layer level, and so only projection and interpolation weights contribute to the drift.

### 3.3.2 GRADIENT DIVERSITY PRESERVATION.

InterpLayers preserve gradient diversity by altering the structure of the NTK. Given the InterpLayer formulation, the gradient with respect to network parameters decomposes as

$$\nabla_\theta h(x) = \begin{bmatrix} z(x) \odot \nabla_{\theta_p} h_{\text{proj}}(x) \\ \nabla_{\theta_z} z(x) \odot \left( h_{\text{proj}}(x) - h_{\text{ref}}(x) \right) \end{bmatrix}. \tag{10}$$

This yields an NTK of the form

$$N_{\text{IL}}(x_i, x_j) = (z(x_i) \odot z(x_j))^\top N_{\text{proj}}(x_i, x_j) + N_{\text{interp}}(x_i, x_j), \tag{11}$$

where $N_{\text{proj}}$ and $N_{\text{interp}}$ denote the NTK contributions from projection and interpolation parameters, respectively. Here $i, j$ index input samples $x_i, x_j$ rather than parameters. Intuitively, the interpolation mechanism adds a persistent gradient component even when the projection pathway degenerates, sustaining diversity in the NTK. For readers unfamiliar with NTK calculations, we provide a step-by-step derivation and empirical estimator details in Appendix A.2.1.

**Theorem 2** (NTK Rank Preservation under Interpolation Variance). *Suppose the interpolation weights $z(x)$ across samples have non-zero variance along at least one coordinate, i.e.,*

$$\text{Var}[z_{(k)}(x)] > 0 \quad \textit{for some dimension } k.$$

*Then the effective NTK rank of an InterpLayer is lower-bounded by*

$$\text{rank}(N_{IL}) \geq \text{rank}(N_{interp}).$$

*In particular, the interpolation pathway guarantees a persistent gradient component, preventing full rank collapse even if the projection pathway degenerates.*

The key requirement of Theorem 2 is simply that interpolation weights exhibit variance across samples. Intuitively, as long as $z(x)$ does not collapse to a constant vector, the interpolation pathway contributes an independent gradient component to the NTK. This guarantees a persistent lower bound on effective rank and prevents full rank collapse, even in cases where the projection pathway degenerates. Empirical verification of NTK rank during training is provided in Appendix J.3.

# 4 RESULTS

## 4.1 EXPERIMENTAL SETUP

We employ the ProcGen environment (Cobbe et al., 2020) to evaluate the proposed framework on CRL settings. As benchmark tasks, we apply three distribution shifts previously introduced by (Juliani & Ash, 2024) on the *Coinrun*, *Jumper*, *Fruitbot*, and *Heist* environments (example visualizations of the shifts are shown in Appendix G). These three variations are named *permute*, *window*, and *expand*. For the *permute* task, at each shift point, we randomly permute all pixels in the observation space. In the *window* task, the random seed used to generate the levels is changed at each shift point. In the *expand* task, training starts with 100 levels, and at each shift point the training set is continuously expanded by increments of 100, ending with 1000 levels after the final shift.

**InterpLayer Baseline.** We choose as our baseline, the architecture where InterpLayers replace the convolutional encoder layers of the policy, and where dropout (Srivastava et al., 2014) is applied to the projection pathway. We name this baseline as **InterpLayers**. Adding dropout aims to increase variance in the projection pathway, which increases the gap between reference and projection. We hypothesize that the characteristics of dropout enhance the effects of our proposed interpolation mechanism. Ablation studies for other InterpLayer variants are presented in Appendix K.

The policy used in the experiments consists of an encoder using 4 convolutional layers followed by a linear layer. The training is performed using PPO (Schulman et al., 2017). Given the additional number of parameters introduced by InterpLayers, we compare it with an architecture using a similar number of parameters as our *standard* baseline. The standard baseline also uses dropout, as its performance is superior to the variant without dropout. Details regarding the training details and computational cost comparison are given in Appendix B and C, respectively. Our method is also compared to two gated architectures, a ResNet-like architecture (He et al., 2016) and Highway Networks (Srivastava et al., 2015). Finally, our method is compared against the top-performing baseline proposed and benchmarked in (Juliani & Ash, 2024): soft shrink-perturb with layer norm (SSP+LN), which mixes the current weight with initialization noise after each optimizer step (check Appendix D for implementation details). The results are average runs of 10 random seeds where training is performed for 50,000 epochs, with distribution shifts being applied every 5,000 epochs.

## 4.2 INTERPLAYER PERFORMANCE UNDER DISTRIBUTION SHIFTS

We evaluate whether InterpLayers can maintain performance across sequential tasks separated by distribution shifts. Fig. 2 shows the normalized performance, defined as the mean reward over the final 50 episodes of each task, normalized relative to the initial task and plotted with shaded regions denoting the standard error across 10 seeds for five network variants: InterpLayers, Highway, ResNet-like, as well as the baselines, Standard with Dropout, and SSP+LN. For all dropout variants, we set the dropout rate to $0.05$.

**Permute**: The permute task involves the most severe shift, forcing full representational relearning. The ResNet-like baseline loses performance after the initial tasks, dropping below $0$ relative to the initial task in all environments. The performance of the Highway Networks baseline also decreases for Coinrun, Heist, and Jumper. Our proposed InterpLayer variant achieved the best performance for Fruitbot and Jumper. InterpLayers and SSP+LN remain above zero and are the top-performing methods for most tasks.

**Window**: Changing to newly generated levels at each shift produces a clear performance separation. InterpLayers and SSP+LN consistently achieve the best performance, while ResNet-like and Highway Networks show some plasticity loss for the four environments. The standard-dropout baseline achieved good performance but did not outperform InterpLayers and SSP+LN.

**Expand**: Increasing the number of levels provides a gradual adaptation challenge. Consistent with the results for permute and window, InterpLayers and SSP+LN achieve the best performance for Coinrun, Fruitbot, and Heist. For Jumper, all methods achieve similar performance curves with final values dropping below $0$ for later tasks. This suggests that generalization is harder in this task.

Across all shift types, overall InterpLayers networks outperform the standard and gated baselines. Compared to SSP+LN, InterpLayers preserve plasticity while requiring less computation (Appendix

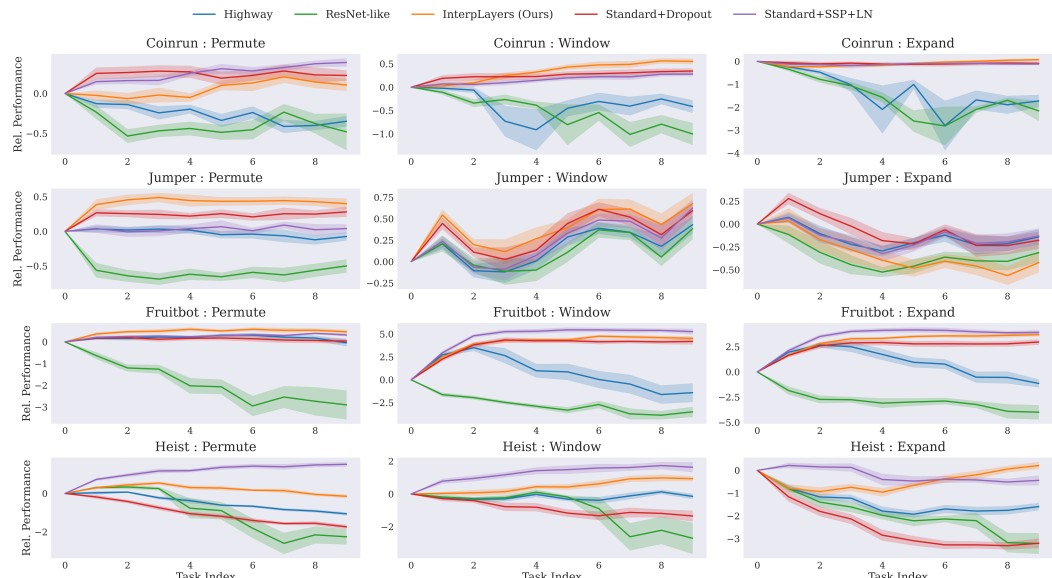

Figure 2: **Performance (relative to the initial task) for InterpLayers and baselines under three distribution shifts for four ProcGen environments.** Performance is defined as the mean reward over the final 50 episodes of each task, normalized relative to the initial task, with shaded regions denoting standard error across 10 seeds.

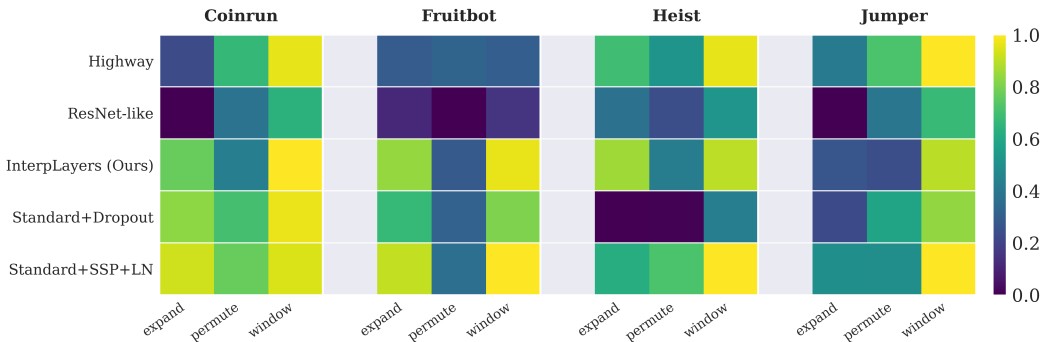

Figure 3: **Raw Rewards (normalized for each task) for InterpLayers and baselines under three distribution shifts for four ProcGen environments at the end of the training.** Across all shift types, InterpLayers networks outperform the standard and gated baselines.

C) and not applying optimization-level interventions. We also evaluate the raw rewards performance in Fig. 3. These results suggest that even though InterpLayers require learning additional interpolation mechanisms, over time, it achieves similar performance to SSP+LN in terms of raw reward performance.

## 4.3 EMPIRICAL VALIDATION OF THEORETICAL PROPERTIES

We show the empirical validation for churn in Fig. 4 for the Coinrun environment. Details on the methodology for calculating this metric are provided in Appendix F. We observe that InterpLayers achieve low churn and reduce churn over time. Adding dropout to a standard baseline is also effective in reducing churn, suggesting that dropout is effective in slowing down plasticity loss. The highest churn values are obtained for the ResNet-like and Highway Networks. SSP+LN maintains a stable churn during the entire training. We observe consistent patterns across all distribution shifts.

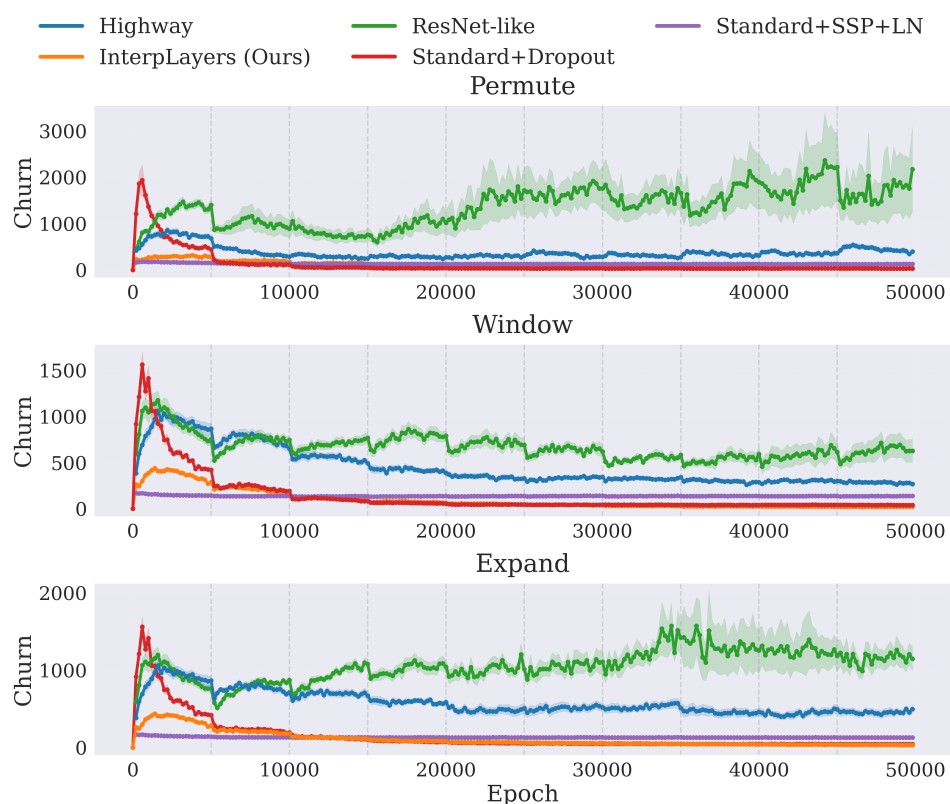

Figure 4: **Evolution of churn for InterpLayers and baselines under distribution shifts in Coinrun.** Shaded regions denote variability across 10 seeds, and vertical lines indicate shift points. Across all conditions, InterpLayers maintain lower churn compared to the standard baselines.

### 4.4 ANALYZING THE INTERPOLATION MECHANISM

Fig. 5 shows per-layer distributions of interpolation weights averaged for early training (tasks 1-5) and late training (tasks 6-10). It is observed that early training is characterized by high variance and broad weight distributions. In late training, the distributions shift towards the reference pathway, which indicates that low-level features are stabilized. We see that the average value for interpolation weights saturates around 0.2. This pattern is more prevalent in the expand and window tasks, while less prevalent in the permute task.

## 5 DISCUSSION

The analysis in Fig. 5 shows that InterpLayers develop a hierarchical structure implicitly. While fixed interpolation weights of $z = 0.5$ would act like ResNet-like skip connections, we observe a different pattern. Across all task shifts, the interpolation weights do not saturate towards 0, 0.5, or 1, but instead settle around $z \approx 0.2$, which indicates a preference for the reference pathway. We find that this pattern happens more consistently in the expand and window shifts, while permute has a slightly higher average value for $z$. This splitting of jobs is not hard-wired into the architecture but develops naturally from the input-specific interpolation. Such self-organization is similar to other findings in deep learning, where lower layers act as feature extractors while higher layers adapt to task-specific demands (Yosinski et al., 2014).

The evolution of metrics related to the theoretical properties presented in Section 3.3 is crucial to mitigate plasticity loss. Our empirical results for churn evolution (Fig. 4) show that it decreases over time using InterpLayers. These results agree with results recently presented by (Tang et al., 2025),

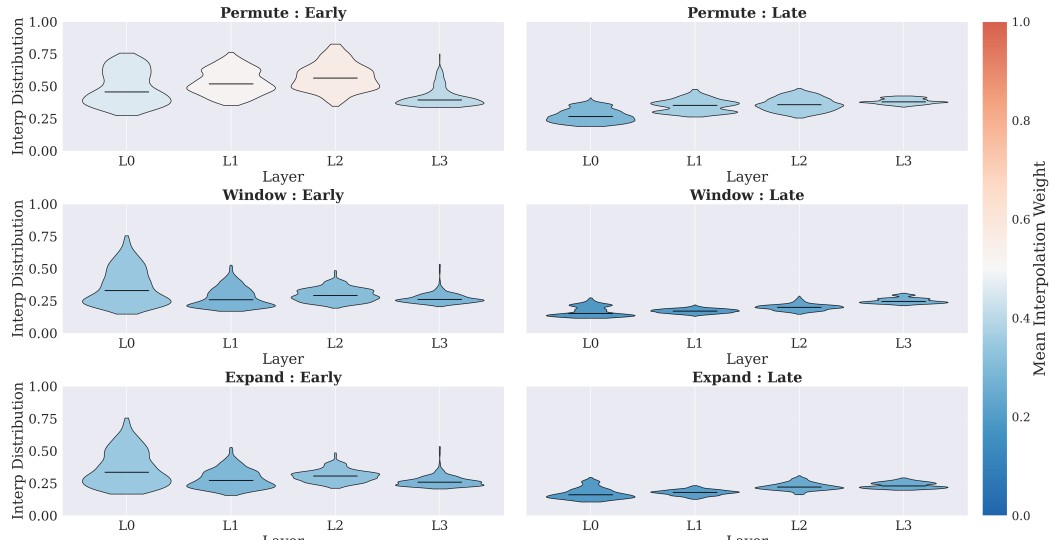

Figure 5: **Distribution per-layer of interpolation weights in early training (tasks 1-5) and late training (tasks 6-10).** Interpolation weights are initialized with a mean equal to 0.5. We can see that at the end of the training, these weights saturate with values around 0.2, with permute achieving higher average values when compared to window and expand.

demonstrating that reducing churn is important to keep plasticity in neural networks. These findings suggest empirically the theoretical advantages of using InterpLayers in continual learning.

Furthermore, the analysis in Section 3.3 suggests that the plasticity of the network can be estimated through the joint behavior of the interpolation weights $z$ and the representational gap $D$ defined in Theorem 1. Together, $z$ and $D$ indicate how much a layer adapts. These variables are important to understand the learning of InterpLayer variants combined with dropout. Dropout is stochastically masking projection activations, preventing projection and reference from aligning, i.e., sustaining $D$. We further discuss how InterpLayers and dropout interact in Appendix H.

Architectures with gated mechanisms (Hochreiter & Schmidhuber, 1997; Cho et al., 2014) and residual networks (He et al., 2016; Srivastava et al., 2015) have been responsible for key advances in recurrent neural networks and convolutional neural networks, respectively. In the same direction, InterpLayers present an interpolation mechanism that sustains plasticity through different streams and gated interventions while also providing a complementary architectural axis to other methods, preventing plasticity loss. This resonates with neuroscience-inspired models where dendritic compartments and gating mechanisms solve the stability-plasticity trade-off in cortical circuits (Bengio et al., 2015; Urbanczik & Senn, 2014). Our findings place InterpLayers as a simple but general mechanism that enriches the toolbox of CRL toward architectures implicitly solving the plasticity loss issue.

## 6 CONCLUSION

In this paper, we introduce InterpLayers as an architectural solution to plasticity loss in CRL. Requiring no schedule, resets, or auxiliary objectives, InterpLayers provide continuous regulation of plasticity through a dual-pathway design. Our findings show that InterpLayers mitigate plasticity loss across four ProcGen environments. We show that combining InterpLayers with dropout improves its performance, achieving comparable performance to state-of-the-art methods for continual learning, suggesting that characteristics learned by dropout regularization help the interpolation dynamics of InterpLayers. Future works include investigating the performance of InterpLayers with different levels of sparsity in the policy network and the combination with different algorithmic approaches in CRL.

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

# A   THEORETICAL PROPERTIES: PROOFS AND EXTENSIONS

## A.1   PROOF OF THEOREM 1

Starting from the first-order output change (Eq. 8):

$$\Delta h(\mathbf{x}) = z(\mathbf{x}) \odot \Delta h_{\text{proj}}(\mathbf{x}) + \Delta z(\mathbf{x}) \odot [h_{\text{proj}}(\mathbf{x}) - h_{\text{ref}}(\mathbf{x})]. \tag{12}$$

By the triangle inequality and the property $\|a \odot b\|_2 \leq \|a\|_\infty \|b\|_2$:

$$\|\Delta h(\mathbf{x})\|_2 \leq \|z(\mathbf{x})\|_\infty \|\Delta h_{\text{proj}}(\mathbf{x})\|_2 + \|\Delta z(\mathbf{x})\|_\infty \cdot D(\mathbf{x}), \tag{13}$$

where $D(\mathbf{x}) = \|h_{\text{proj}}(\mathbf{x}) - h_{\text{ref}}(\mathbf{x})\|_2$.

By Lipschitz continuity assumptions:

$$\|\Delta h_{\text{proj}}(\mathbf{x})\|_2 \leq L_p \|\Delta \theta_p\|_2, \tag{14}$$

$$\|\Delta z(\mathbf{x})\|_\infty \leq L_z \|\Delta \theta_z\|_2. \tag{15}$$

Therefore:

$$\|\Delta h(\mathbf{x})\|_2 \leq \|z(\mathbf{x})\|_\infty L_p \|\Delta \theta_p\|_2 + L_z \|\Delta \theta_z\|_2 D(\mathbf{x}). \tag{16}$$

Since $z(\mathbf{x}) \in (0,1)^h$ due to the sigmoid, $\|z(\mathbf{x})\|_\infty < 1$, completing the proof. $\qquad\square$

## A.2   COROLLARY: BOUNDED CHURN

Consider a sequence of updates $\{\theta_t\}_{t=0}^T$ under learning rate $\eta$. By Theorem 1, each step incurs an output change bounded by

$$\|\Delta h_t(\mathbf{x})\|_2 \leq \eta \left( \|z(\mathbf{x})\|_\infty L_p \|\nabla_{\theta_p} \mathcal{L}_t\|_2 + L_z \|\nabla_{\theta_z} \mathcal{L}_t\|_2 D(\mathbf{x}) \right). \tag{17}$$

Accumulating over $t$ and applying Cauchy–Schwarz yields

$$\|h_{\theta_T}(\mathbf{x}) - h_{\theta_0}(\mathbf{x})\|_2 \leq BT, \tag{18}$$

for a constant $B$ depending on $\eta$, $L_p$, $L_z$, and the gradient magnitudes. Squaring and taking expectation over $\mathcal{D}_{\text{ref}}$ gives

$$\mathcal{C}_T \leq B^2 T^2, \tag{19}$$

establishing bounded churn.

### A.2.1   PROOF OF THEOREM 2

We restate the NTK for InterpLayers (Eq. 11):

$$N_{\text{IL}}(x_i, x_j) = (z(x_i) \odot z(x_j))^\top N_{\text{proj}}(x_i, x_j) + N_{\text{interp}}(x_i, x_j). \tag{}$$

**Step 1: PSD decomposition.**   Both $N_{\text{proj}}$ and $N_{\text{interp}}$ are positive semidefinite (PSD) Gram matrices of gradients. Therefore, their weighted sum is also PSD. The interpolation kernel can be written explicitly as

$$N_{\text{interp}}(x_i, x_j) = \left\langle \nabla_{\theta_z} z(x_i) \odot (h_{\text{proj}}(x_i) - h_{\text{ref}}(x_i)), \ \nabla_{\theta_z} z(x_j) \odot (h_{\text{proj}}(x_j) - h_{\text{ref}}(x_j)) \right\rangle,$$

which is PSD by construction.

**Step 2: Rank contribution of interpolation.**   If $z(x)$ collapses to a constant vector $c$ across all samples, then the interpolation gradients vanish (since $\nabla_{\theta_z} z(x)$ is zero almost everywhere after saturation). In that case $N_{\text{interp}}$ degenerates to zero. Conversely, if $\text{Var}[z_{(k)}(x)] > 0$ for at least one coordinate $k$, then the interpolation gradients differ across samples, producing at least one non-zero eigenvalue in $N_{\text{interp}}$.

**Step 3: Rank inequality.** Because $N_{\mathrm{IL}} = \underbrace{(z_i \odot z_j)^\top N_{\mathrm{proj}}}_{\text{possibly degenerate}} + N_{\mathrm{interp}}$ and both terms are PSD, we

have

$$\mathrm{rank}(N_{\mathrm{IL}}) \geq \mathrm{rank}(N_{\mathrm{interp}}).$$

This follows from the fact that adding a PSD matrix cannot reduce the rank contribution of another PSD component.

**Step 4: Conclusion.** Thus, provided $z(x)$ is not constant across all samples, the interpolation term guarantees a non-zero rank contribution. In particular, $\mathrm{rank}(N_{\mathrm{IL}})$ cannot collapse below $\mathrm{rank}(N_{\mathrm{interp}})$, ensuring gradient diversity even if $N_{\mathrm{proj}}$ degenerates.

$\square$

### A.3 WEIGHT NORM REGULARIZATION (EXTENDED)

Although not central to the main text, we note that interpolation gates implicitly regularize effective weight norms. Define the effective contribution at time $t$ as

$$\|\mathbf{W}_{\mathrm{eff}}(t)\|_F^2 \leq \mathbb{E}_x[\|z_t(x)\|_\infty^2] \cdot \|\mathbf{W}_{p,t}\|_F^2 + \|\mathbf{W}_{z,t}\|_F^2, \tag{20}$$

Since $\|z_t(x)\|_\infty \leq 1$, the contribution of $\mathbf{W}_{p,t}$ is strictly bounded relative to its norm. This prevents unbounded growth of effective weights even when $\|\mathbf{W}_{p,t}\|_F \to \infty$.

## B TRAINING DETAILS

**Highway Networks.** We implement this baseline following Srivastava et al. (2015), where each layer computes an interpolation between a non-linear transformation and a skip pathway. For any input $x$, the network computes

$$y = T(x) \odot H(x) + (1 - T(x)) \odot C(x) \tag{21}$$

where $H$ is a learnable nonlinear transformation, $T = \sigma(\mathrm{gate}(x))$ is a sigmoid gate, and $C$ is the skip pathway. $H$ is either a linear or convolution layer, followed by a ReLU non-linear activation function. The gate $T$ is given by a parallel linear or convolution layer with its bias initialized to a negative value ($b = -1$). If input and output dimensions differ, we use a learnable projection in the skip pathway to match shapes. We use the same encoder structure described in Section 4.1 in all network variants. Each highway layer creates a transform output, skip output, and gate values, as all pathways are fully learnable.

**ResNet-like Network.** We implement the ResNet-like baseline following He et al. (2016). Each block consists of two convolutional layers with an identity skip connection. For any input $x$, the block computes

$$y = \phi(Conv_2(Conv_1(x)) + C(x)), \tag{22}$$

where $\phi$ is a RELU non-linear activation function and $C(x)$ is the skip pathway. If channel dimensions or spatial resolution change (stride $> 1$), we use a $1 \times 1$ projection that aligns the skip dimensions with the residual one. Following He et al. (2016), we initialize the second Conv-block to zero in order to ensure that the block behaves as an approximate identity mapping.

**Training protocol.** The RL policy is trained using Proximal Policy Optimization (PPO) (Schulman et al., 2017). Following the framework described in (Juliani & Ash, 2024), we report the performance at the epoch level and mark task boundaries at each distribution shift. In our setup, one epoch denotes the following steps: (i) we collect buffer_size = 1024 transitions across 11 parallel environments, then (ii) perform 3 PPO passes with minibatch size set to 64. The PPO hyperparameters are set as follows: $\gamma = 0.99, \lambda = 0.95$, clip = 0.2, entropy = 0.02, learning rate = 5e-4. We train the policy for 50,000 epochs, with distribution shifts at every 5000 epochs, i.e., 5000, 10000, ..., 45000.

## C    COMPUTATIONAL COST COMPARISON

Table 1 presents the parameter counts and forward-pass FLOPs for the main architectures evaluated in this paper. We count one multiple-accumulate as a single FLOP. The conv128 encoder requires nearly 35% more computational load than the InterpConv64 variant used in our InterpLayers, despite the latter showing higher performance in later experiments.

| Encoder Variant | Params (M) | FLOPs (M) |
|---|---|---|
| Conv128 (standard) | 1.98 | 63.5 |
| Conv128 (standard+SSP+LN) | 1.98 | 67.5 |
| InterpConv64 (fullinterp) | 1.52 | 50.8 |
| InterpConv64 (convonly) | 0.99 | 49.7 |

Table 1: Parameter counts and forward FLOPs per inference step.

Table 2 displays the wall-clock training time and memory usage for each of the five conditions we evaluated in Fig. 2.

| Condition | # Runs | Avg Time/Run (hrs) | Avg Final Memory (MB) |
|---|---|---|---|
| Highway | 120 | 23.65 | 8265.8 |
| ResNet-like | 120 | 21.55 | 8318.2 |
| Standard+SSP+LN | 120 | 23.71 | 8309.6 |
| Standard+Dropout | 120 | 21.68 | 8305.2 |
| InterpLayers (Ours) | 120 | 25.78 | 8311.8 |

Table 2: Wall-clock training costs across all experiments.

## D    SOFT SHRINK-PERTURB WITH LAYERNORM (SSP+LN)

We implement soft shrink-perturb following (Juliani & Ash, 2024), where after each optimizer step we apply the shrink and perturb update to the parameters $x$:

$$x_{\text{new}} = \alpha\, x_{\text{current}} + \beta\, x_{\text{init}}, \quad x_{\text{init}} \sim \mathcal{D}_{\text{init}}. \tag{23}$$

with $\alpha = 0.999999$ and $\beta = 0.000001$

In SSP+LN, this continuous update is combined with LayerNorm (Ba et al., 2016) applied throughout training.

## E    DETAILS ON THE NTK COMPUTATION

We measure the empirical NTK of the policy and value PPO heads throughout training. The goal is to understand if InterpLayers maintain gradient diversity compared to baselines.

**Reference batch.**    At initialization, we collect a reference batch of observations from the training environments. To ensure diversity, samples are drawn from multiple environments using three strategies: (i) fresh resets, (ii) short random walks, and (iii) mid-episode states. We target 200 samples, capped at 50 per environment. If fewer samples are available, we fall back to a minimum of 16.

**NTK matrix construction.**    For each reference observation $x$, we compute the gradient of the PPO objective with respect to all trainable parameters of the policy (and optionally value) networks:

$$g(x) = \nabla_\theta \mathcal{L}_{\text{PPO}}(x).$$

The empirical NTK matrix is then

$$K_{ij} = \langle g(x_i), g(x_j) \rangle.$$

Gradients are computed in mini-batches, and the resulting kernel is assembled as a Gram matrix of dimension up to $200 \times 200$.

**Effective rank and spectra.** We report the *effective rank* of the NTK, defined as the participation ratio:

$$r_{\text{eff}} = \frac{\left(\sum_k \lambda_k\right)^2}{\sum_k \lambda_k^2},$$

where $\lambda_k$ are eigenvalues of $K$. This value measures the number of significant gradient directions. We also record the minimum eigenvalue and condition number to diagnose degeneracy.

**Logging frequency and cost.** NTK metrics are computed every 250 epochs, aligned with test evaluations. Each computation uses the fixed reference batch from initialization and incurs approximately 10–15% additional runtime overhead relative to standard PPO training.

**Implementation.** The NTK logger is implemented in PyTorch and integrated into the PPO training loop. It automatically detects whether gating is enabled and saves all metrics to disk in JSON/CSV format for post-hoc analysis.

# F    DETAILS ON THE CHURN COMPUTATION

We measure churn from the encoder outputs using a fixed reference batch that is stored at initialization. At epoch $t$, churn is defined as the mean squared deviation of the current encoder representations from the initial ones:

$$\mathcal{C}_t = \mathbb{E}_{x \sim \mathcal{D}} \left[ \| h_t(x) - h_0(x) \|_2^2 \right], \tag{24}$$

where $h_t(x)$ denotes the encoder representation of input $x$ at epoch $t$, and $h_0(x)$ the corresponding representation at initialization. We also log the first- and second-order finite differences of $C_t$ over epochs.

# G    VISUALIZATION OF THE DISTRIBUTION SHIFTS OF PROCGEN TASKS

Sample visualizations for three ProcGen coinrun tasks evaluated in this paper are shown in Figure 6. For **permute**, a fixed random pixel permutation is applied per shift. Given the change in the entire state space, this task requires robust feature relearning. For **window**, the environment is resampled with a different random seed to create other environments. Finally, the expand tasks increase the number of training environments from 100 to 1000 across 9 shifts. This characteristic evaluates the generalization capabilities of the trained policy.

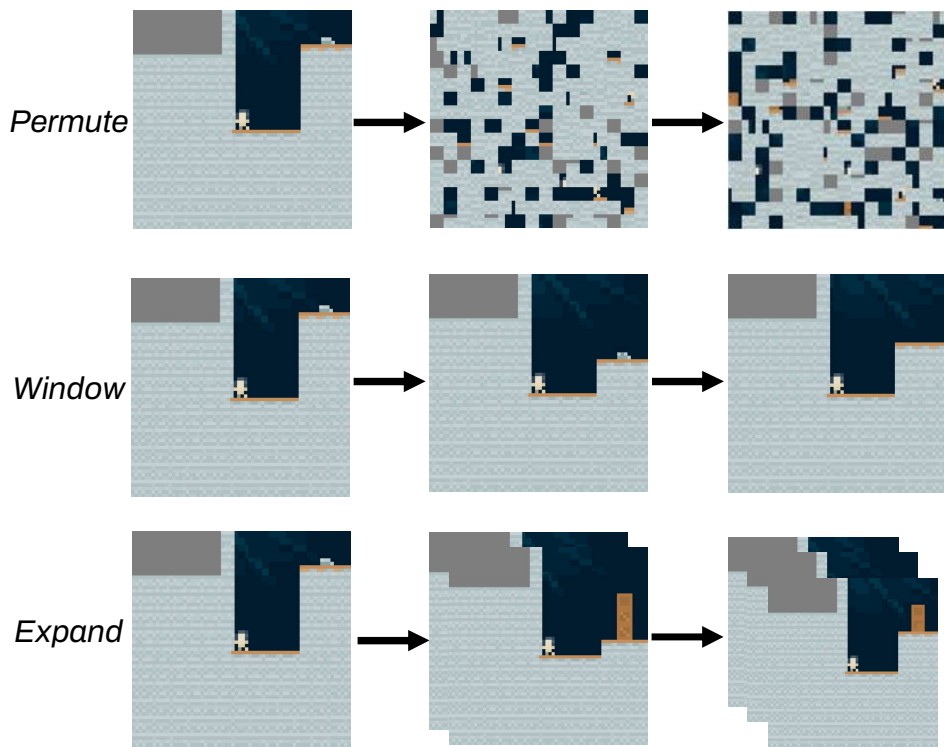

Figure 6: **Visualization of the distribution shifts used in ProcGen coinrun.** Each panel shows the transformation applied at the shift boundaries (every 5,000 epochs): **Permute** applies a fixed random pixel permutation per shift; **Window** resamples the environment seed to generate new levels; **Expand** increases the number of training environments from 100 to 1000 across 9 shifts. Visual representations of the environments are shown.

## H    EXTENDED DISCUSSION

### H.1    WHY DROPOUT IMPROVES INTERPLAYER RESULTS

Different from the standard application in neural networks, where dropout masks neurons in all layers, in our method, we only apply dropout to the projection pathway. Our intuition is that applying dropout only to the projection increases the activation variance of the projection pathway, affecting the representational gap $D$ defined in Theorem 1. In Theorem 1, we show that InterpLayers provide an upper bound to their output variability by:

$$\|\Delta h(x)\|_2 \ \leq \ \|z(x)\|_\infty \, L_p \, \|\Delta\theta_p\|_2 \ + \ L_z \, \|\Delta\theta_z\|_2 \, D(x), \qquad D(x) = \|h_{\mathrm{proj}}(x) - h_{\mathrm{ref}}(x)\|_2. \tag{25}$$

In this equation, the term $D$ determines how much changes in the interpolation weights affect the output. In this case, if the projection and reference are equal, the interpolation weights do not have an impact on the output. However, if $D$ increases, the impact of the interpolation pathway also increases. Dropout strictly increases the activation variance in the projection pathway by injecting noise. So the instantaneous gap is as follows

$$D_t(x) = \left\|\tilde{h}_{\mathrm{proj},t}(x) - h_{\mathrm{ref}}(x)\right\|. \tag{26}$$

This way, dropout guarantees that $D_t(x)$ always has some variance, stabilizing the gradient of the interpolation pathway $z(x)$ in Equations 4 and 7.

**InterpLayers with Dropout vs Standard Networks with Dropout.** In Figure 2, we compare InterpLayers with standard networks with dropout. We show that applying dropout also improves the plasticity of standard networks significantly. However, it does not fully prevent plasticity loss across all conditions, and it does not outperform InterpLayers. We hypothesize that this difference in performance happens because standard layers do not benefit from the additional activation variance as InterpLayers, which contain both a reference and a projection pathway. Instead, for standard networks, dropout decreases model-level variance, which slows down the performance collapse but does not prevent it, as the empirical results suggest.

### H.2 PLASTICITY INDEX

Following the intuition from the previous section, we list two independent mechanisms that control plasticity in InterpLayers:

1. The representational gap $D$

$$D(x) = \|h_{\mathrm{proj}}(x) - h_{\mathrm{ref}}(x)\|_2.$$

   High values for $D$ mean that even small changes to the interpolation weights yield large changes in the output.

2. The interpolation weights $z$
   If these weights are close to 0 (or 1, for that matter), the magnitude of $D$ becomes less impactful on the final output.

When combined, $z$ can be interpreted as a "exposure" term to the projection pathway and $D$ as a "sensitivity" term to that exposure. Combining them quantifies how "plastic" a layer is, which we define as a plasticity index:

$$PI(x) = z(x)\, D(x).$$

If $PI = 0$, the layer shows no plasticity because either the projection has no influence on the output (if $z = 0$) or because the reference and projection are indistinguishable (if $D = 0$). $PI > 0$ means that the layer is somewhat plastic because the projection is different from the reference, and the interpolation exposes that difference. Dropout affects this index due to its "variance injection" (if we want to follow the common term of plasticity injection). This variance ensures that even if $D$ is overall decreasing, the instantaneous $D_t$ will show some fluctuations. As the interpolation weights' gradients are dependent on $D$, this means that they would never become dormant. This maintains $PI$ to be non-zero.

### H.3 INTERPOLATION DISTRIBUTION MAINTAIN VARIANCE

Figure 5 show that the interpolation weights do settle around $0.2$ across all tasks. However, this illustration alone does not provide enough information about whether they maintain their variance, which is a critical part to guarantee stability according to Theorem 2. To evaluate their variance, we compute the Normalized Gate Diversity Ratio (NGDR) for each layer

$$NGDR(t) = \frac{Var[z_t]}{\mu_t(1 - \mu_t)} \tag{27}$$

where $z_t$ denotes the interpolation weights of a layer at epoch $t$ and $\mu_t$ is their mean. The denominator is the variance of a Bernoulli distribution with mean $\mu_t$, so that the NGDR works as a scale-free measure of how diverse the interpolation weights are relative to the maximal possible variance. Figure 8 shows that across all shifts in all tasks, the interpolation values saturate toward values around $0.2$ (as mentioned above), whereas their NGDR remains stable (between $0.3$ and $0.5$) during training. This suggests that the interpolation weights do not collapse to a single value but instead remain or even increase their variance. This empirically supports the mechanism behind Theorem 2: that interpolation distributions maintain non-degenerate variance, which consequently preserves gradient diversity and a stable effective NTK rank.

# I GUIDELINES FOR CHOOSING INTERPLAYER VARIANTS

InterpLayers can be applied to any layer in a neural network. In our ProcGen environment, the network architecture is an encoder followed by respective PPO heads, following Juliani & Ash (2024). The encoder consists of four convolutional layers and one linear layer. The convolutional stack acts as a feature encoder, with the linear layers combining these features accordingly. In this work, we evaluate two InterpLayer variants:

$$\textbf{convonly} : \text{InterpLayer is only applied to the convolutional layers of the encoder.} \quad (28)$$
$$\textbf{fullinterp} : \text{InterpLayer is applied to all layers of the encoder.} \quad (29)$$

Omitting the InterpLayer from the linear layer reduces the parameter count by 524,544. This difference is significant if memory or throughput are limiting factors. Across all three task shifts, the convonly variant performs equal to or better than the fullinterp variant, especially when combined with dropout. This suggests that most of the benefit of interpolation occurs in the convolutional part of the encoder. Thus, we recommend using the **convonly** variant when the architecture has a clear feature encoder or computational efficiency is important, and to use **fullinterp** in scenarios where there are no computational restrictions or the training is unstable after task changes.

# J EXTENDED RESULTS

## J.1 RAW RESULTS

We show the raw returns graph obtained during training in Fig. 7. It is seen that SSP+LN consistently achieves the highest rewards. InterpLayers achieve good performance, especially for the window and expand tasks. It is interesting to observe that, even though Highway Networks show plasticity loss in different scenarios, they achieve convergence speed and raw reward similar to SSP+LN for the first tasks.

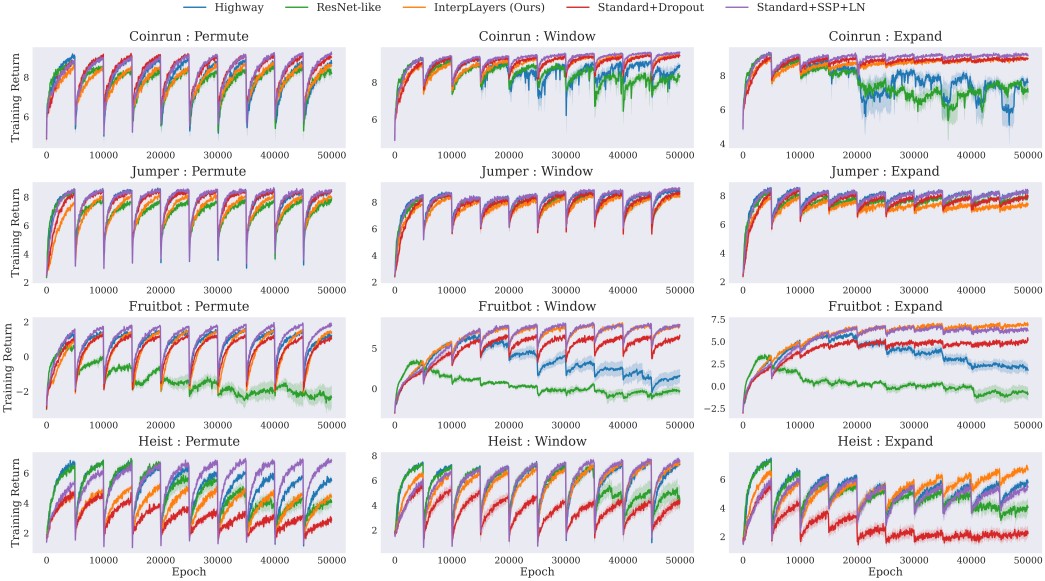

Figure 7: **Raw rewards obtained during training for the methods evaluated.**

## J.2 INFLUENCE OF DROPOUT RATE

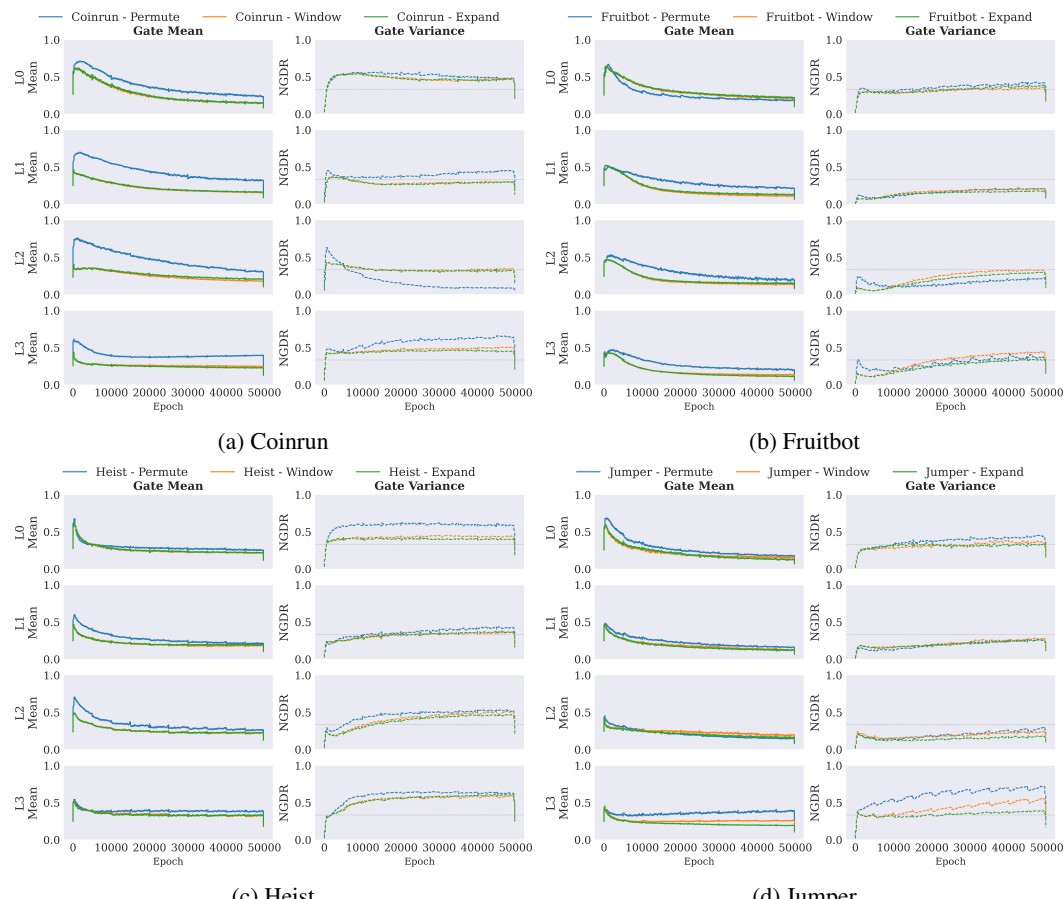

(a) Coinrun

(b) Fruitbot

(c) Heist

(d) Jumper

Figure 8: **Evolution of the layer-wise gate means and NGDR across tasks and shift types.** The interpolation means settle around $\approx 0.2$, while the NGDR remains stable between $0.3$ and $0.5$ even in late training stages. This indicates that the gate variance does not collapse which supports the theoretical claim in Theorem 2 that InterpLayers maintain gradient diversity thus a stable NTK rank.

We conduct a small ablation study in the Coinrun environment using different dropout rates ranging from no dropout to a dropout rate of $0.2$ for 6 seeds each. Figure 9 shows the performance retention and raw returns of different variants for the three distribution shifts. It is observed that higher dropout rates lead to minimally higher relative performance, however, at the cost of losing raw reward performance. We can also observe that no dropout, while having very high initial performance, shows strong performance degradation, especially in the expand condition. Therefore, it is important to choose a dropout rate that keeps plasticity while being still able to solve the environments where the policy will be applied. In this work, we choose a dropout rate of $0.05$ for the default InterpLayer variant.

## J.3 NTK Analysis

In Theorem 2, we outline theoretically how InterpLayers have properties that prevent the NTK rank from collapsing. Here, we measure NTK metrics following the methodology described in Appendix E. Due to computational constraints, we measure the empirical NTK only in the PPO heads instead of throughout the entire network. Fig. 10 shows the effective NTK rank of InterpLayers and the four baselines in the Fruitbot environment. The results show that all variants maintain their effective rank even in late training. This analysis is still limited in scope, and future work should explore whether there are significant differences in the gradients of the encoder. Additionally, if these patterns cor-

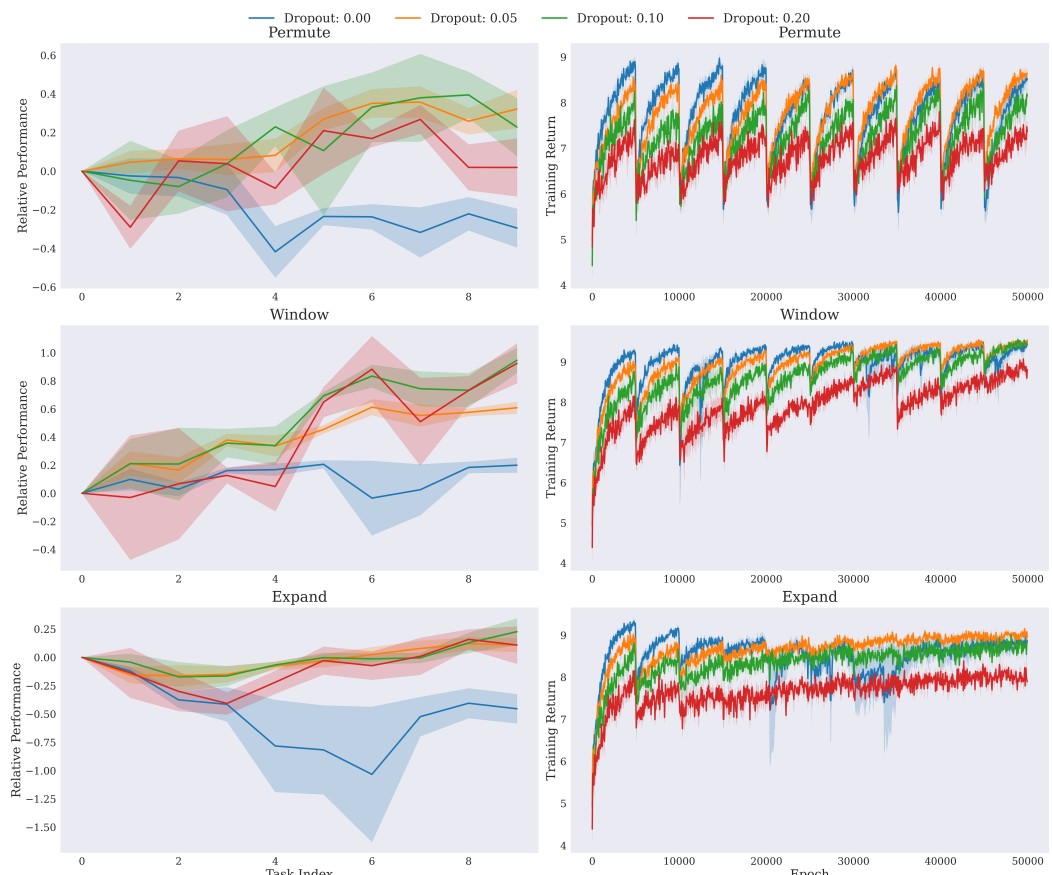

Figure 9: **Ablation study for the dropout rates applied to the projection-pathway.** Higher dropout rates show minimally better relative performance but show strong decreases in raw returns, while removing dropout (0.00) shows substantial performance degradation, especially in the expand shift. The dropout rate of 0.05 shows the best balance between performance retention and raw performance.

relate with mitigating plasticity loss during extended training periods in continual learning settings.

## K    COMPARISON OF INTERPLAYERS VARIANTS

We evaluate two architectural variants: (i) **convonly**, where InterpLayers replace only the convolutional encoder layers, and (ii) **fullinterp**, where both convolutional and linear layers are replaced with InterpLayers. We also investigate InterpLayers combined with dropout (Srivastava et al., 2014), which we name **convonly-dropout** and **fullinterp-dropout** respectively, in which dropout is applied to the projection pathway. The convonly variant emphasizes stability in low-level feature extraction, while fullinterp exposes the entire network to interpolation.

We compare InterpLayer variants in the Coinrun environment using 10 random seeds for each condition in each shift. Figure 11 shows that the non-dropout variants outperform the dropout ones in terms of raw performance in the permute condition, but they show significant performance degradation in the other two tasks. Between convonly-dropout and fullinterp-dropout variants the perfor-

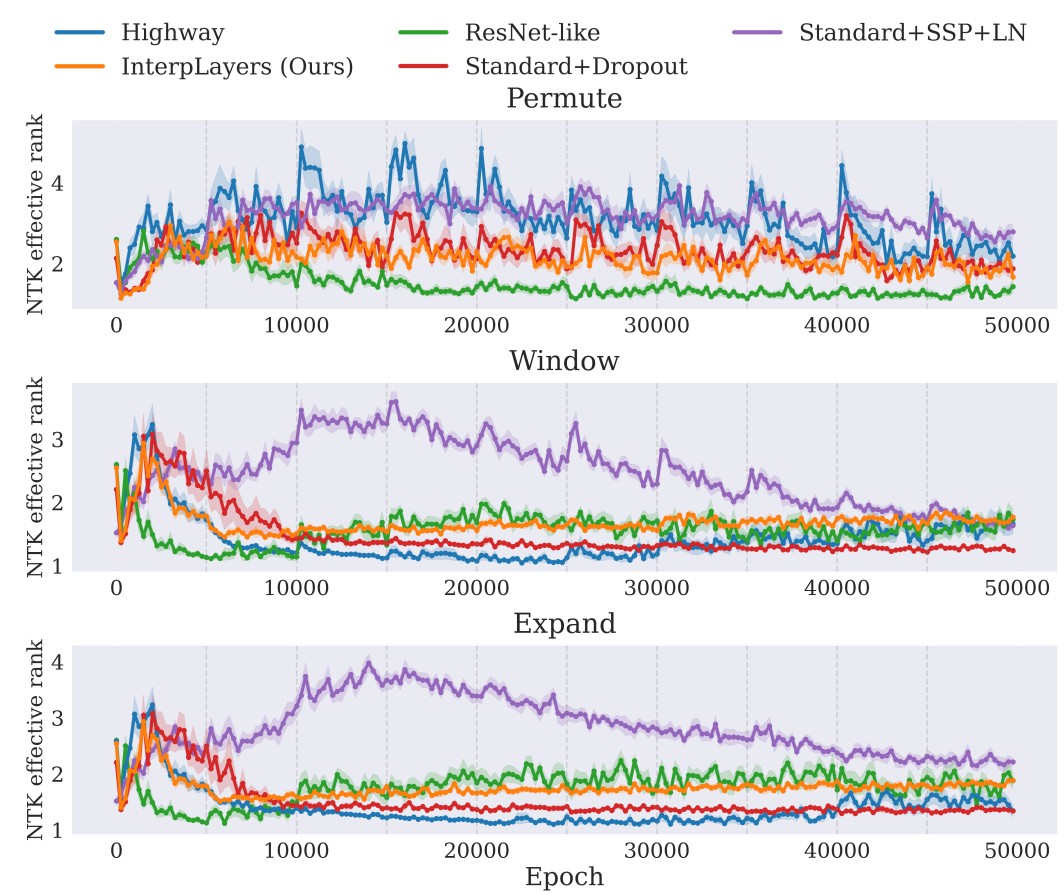

Figure 10: **Evolution of effective NTK rank for InterpLayers and baselines under distribution shifts in Fruitbot.** Shaded regions denote variability across 10 seeds, and vertical lines indicate shift points. Across all conditions, all variants show stable rank besides SSP+LN which first shows a strong increase followed by a gentle decrease.

mance is similar. We chose the **convonly-dropout** as our default variant throughout our experiments as it is computationally cheaper than the fullinterp-dropout variant (see Section C).

## L  ABLATION WITH PERMUTED TASK ORDER

In this section, we aim to verify that InterpLayers are not overfitting on a specific task sequence, i.e., that they learn the pattern of a shift and not actually the newly presented task. For this, we performed all shifts a priori and stored the environments. Then we shuffled the task sequence so that an original sequence of 1-2-...-9 would become, for example, 1-5-...-3. Figure 12 shows that InterpLayers reach the same level of performance retention and raw performance in both settings (original and permuted). In this way, we show that InterpLayers are robust to random task sequences, an important property in continual learning settings.

## M  ORTHOGONALLY COMBINING INTERPLAYERS TO ALGORITHMIC SOLUTIONS TO PLASTICITY LOSS

InterpLayers can serve as an orthogonal solution to algorithmic approaches to mitigate plasticity loss. Here, we combine InterpLayers with LayerNorm (LN) and SSP-LN, and evaluate their per-

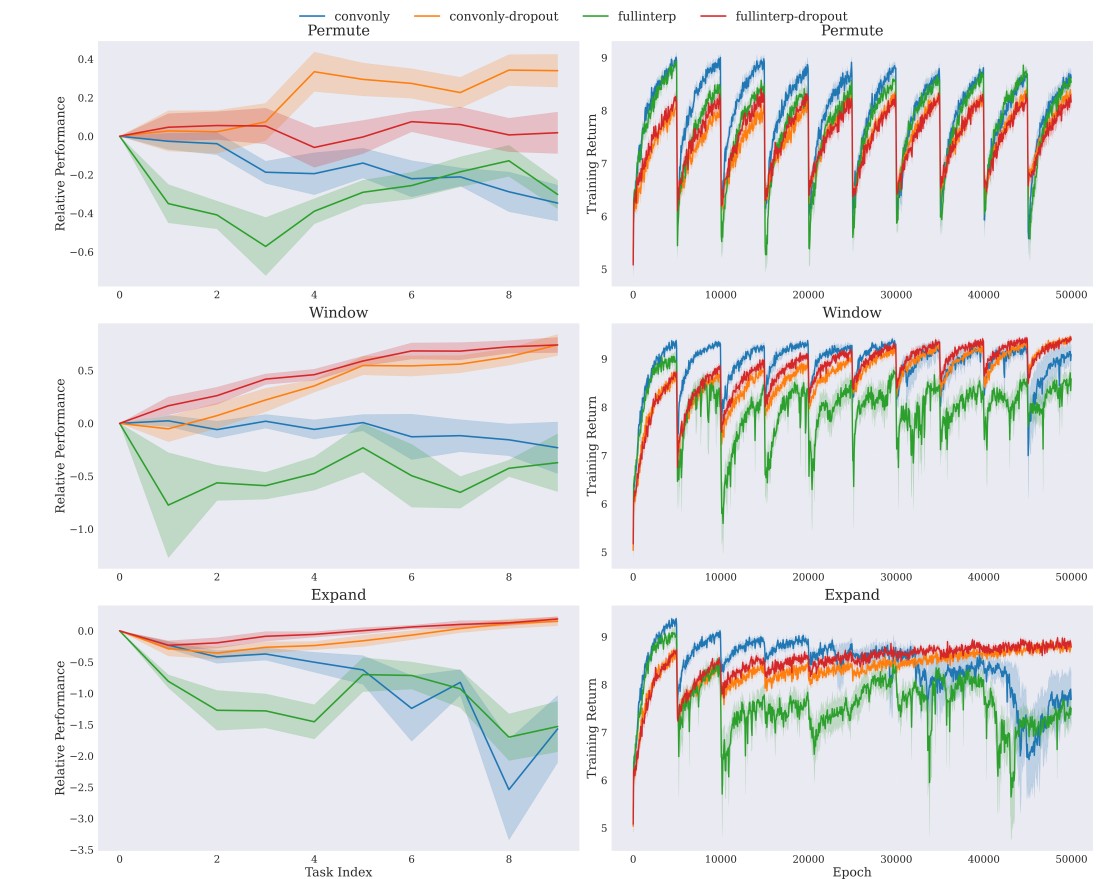

Figure 11: **Comparison of different InterpLayer variants in Coinrun.**

formance for the three distribution shift types across 10 random seeds. For this ablation, we use the convonly-dropout variant. Figure 13 shows that combining InterpLayers with LN improves the initial convergence speed of InterpLayers, enhancing its performance for the first tasks. We also observe that combining InterpLayers with SSP-LN yields the same performance as combining it only with LN.

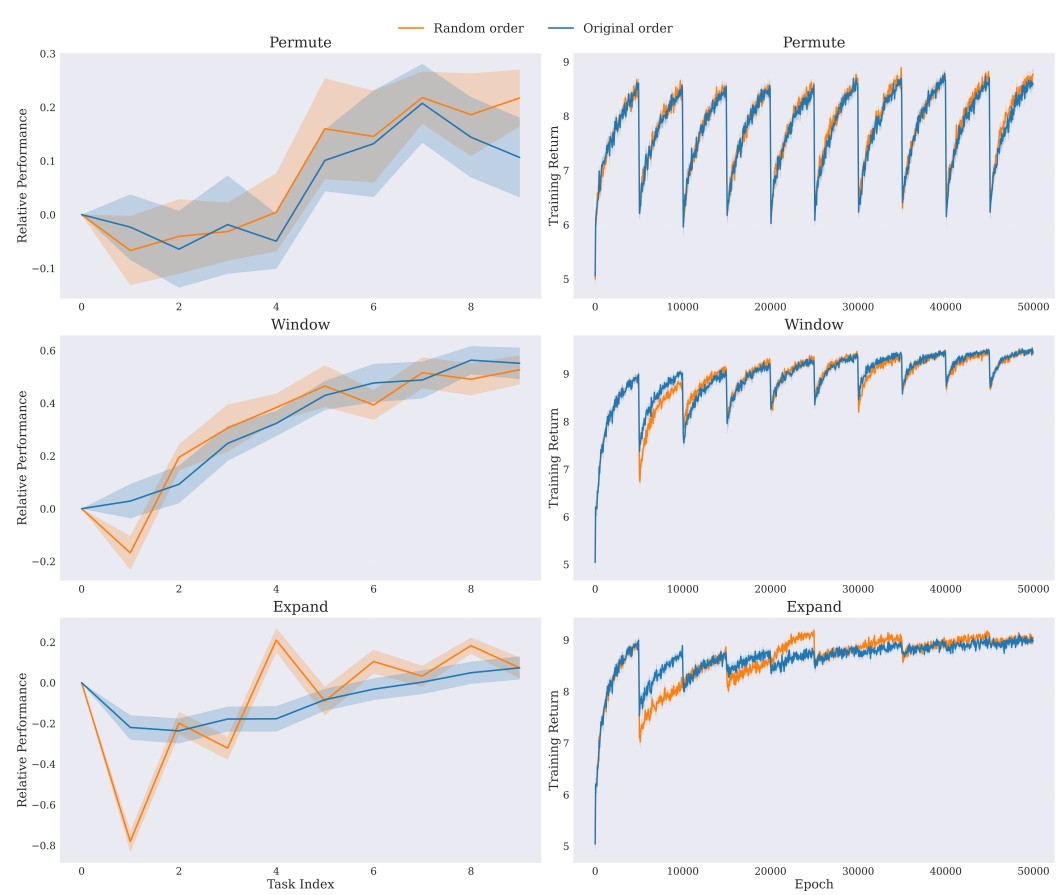

Figure 12: **Results for ablation study with task order permuted.** InterpLayers show consistent results for both settings, showing robustness to randomly permuted tasks.

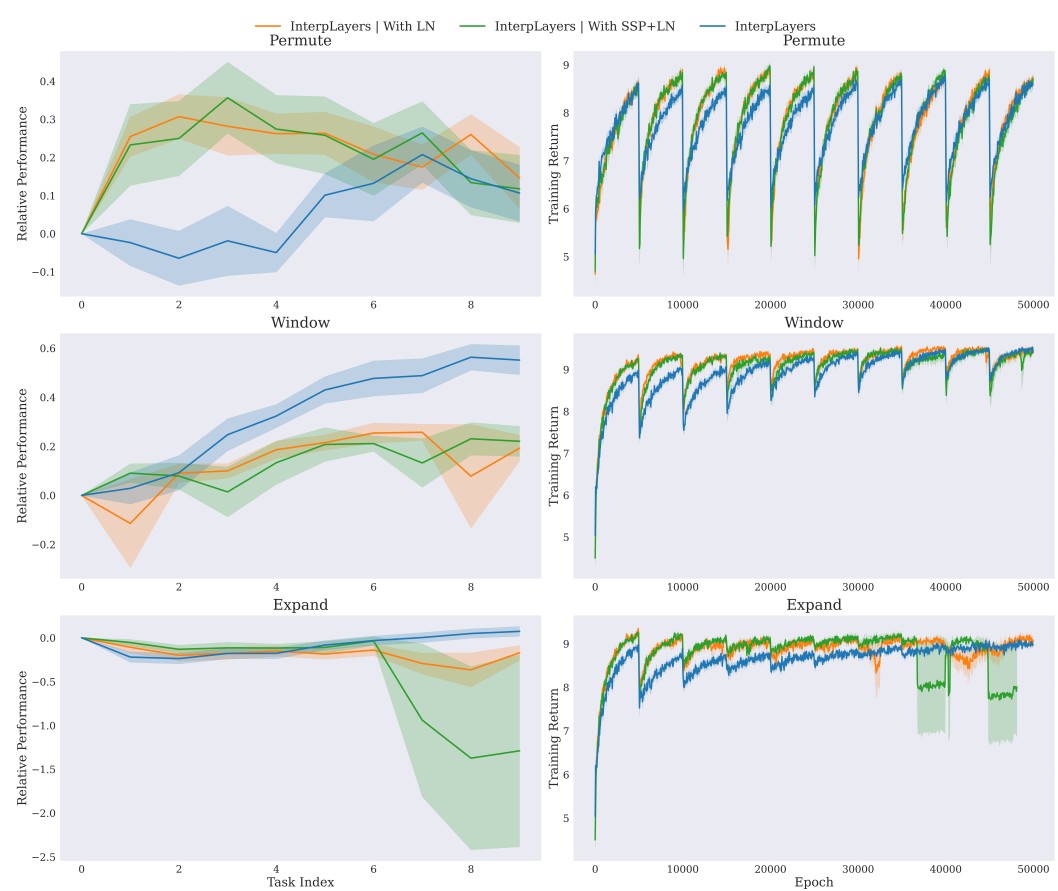

Figure 13: **Ablation study combining InterpLayers with LayerNorm and SSP-LN.** Combining InterpLayers with these methods improves its convergence speed for the initial tasks during training.

