# OpenReview forum: "Architectural Plasticity for Continual Learning"
_ICLR.cc/2026/Conference — Submitted to ICLR 2026_

### Official Review · Reviewer_N1pd · 2025-10-27

**Soundness:** 2
**Presentation:** 2
**Contribution:** 2
**Rating:** 2
**Confidence:** 3

**Summary:**

This work proposes **InterpLayer**, a new neural network layer designed to address the issue of plasticity loss in continual reinforcement learning. The authors provide theoretical analysis of two key properties of the layer: bounded representational drift and preservation of gradient diversity—which offer insight into its stability and adaptability. Finally, empirical studies on the ProcGen benchmark demonstrate the effectiveness of the proposed layer, showing improved plasticity, reduced churn, and better preservation of NTK rank.

**Strengths:**

- InterpLayer can directly replace standard MLP and convolutional layers without further modification or additional hyperparameters, offering a simple enhancement to existing architectures.
- The theoretical properties of InterpLayer are promising and well-grounded.

**Weaknesses:**

- Only one environment (*coinrun*) from the ProcGen benchmark is used in the experiments. This seems insufficient for a comprehensive empirical study. I highly recommend that the authors include additional environments to better support the claimed effectiveness of the proposed layer.
- Although Figure 2 shows that InterpLayer achieves good performance relative to the initial task, the raw returns presented in Figures 6–8 indicate that *Standard+SSP+LN* consistently achieves the best overall performance and outperforms all variants of InterpLayer by a substantial margin. If this is not an mistake, does InterpLayer obtain worse initial task performance when compared to *Standard+SSP+LN*? I also recommend moving the raw return plots from the appendix to the main paper to make this comparison more transparent.

**Questions:**

- Would it be possible for the authors to run additional experiments with a **Standard+SSP+LN+dropout** configuration, as it seems to represent a strong and relevant baseline for comparison?

---

> ### Author Response · Authors · 2025-12-03
>
> The authors are incredibly thankful to the reviewer for your insightful and detailed comments that helped improve our work.
>
> ---
>
> ### Comment 1. (Limited Experimental Evaluation)
> > *I highly recommend that the authors include additional environments to better support the claimed effectiveness of the proposed layer.*
>
> Thank you for your comment. According to the reviewer’s suggestion, we performed the experiments for three more ProcGen environments, Jumper, Fruitbot, and Heist. The results show that for the majority of tasks and shift types, InterpLayers outperform ResNet-like architectures and Highway Networks in terms of mitigating plasticity loss and achieve comparable performance when compared to SSP+LN.
>
> Please check the response to Comment 2 from Reviewer 2fPo and Fig. 2 in the revised manuscript for full results.
>
> ---
>
> ### Comment 2. (Raw Return Performance)
> > *Does InterpLayer obtain worse raw rewards performance when compared to Standard+SSP+LN? I also recommend moving the raw return plots from the appendix to the main paper.*
>
> Thank you for your insightful comment. According to the reviewer’s suggestion, we created a new raw rewards heatmap plot and added it to the main text as Fig. 3. For the raw reward performance of InterpLayers, we observe that the convergence speed is slower for the Permute task but comparable for the Window and Expand tasks when compared to other baselines.
>
> We would like to acknowledge that the goal of our study is to show that architectural choices can be crucial to mitigate plasticity loss. Therefore, following the methodology presented in [1], we focus on performance retention as the main comparison metric for our main analysis. We believe that this novel architectural plasticity research direction can lead to many impactful works in continual learning.
>
> [1] Juliani, Arthur, and Jordan Ash. "A study of plasticity loss in on-policy deep reinforcement learning." Advances in Neural Information Processing Systems (2024).
>
> ---
>
> ### Comment 3. (SSP+LN with Dropout)
> > *Would it be possible to run Standard+SSP+LN+dropout?*
>
> Thank you for your suggestion. We ran an ablation study combining Standard-SSP+LN with Dropout. The results show that this combination has similar performance when compared to Standard-SSP+LN.
>
> ### CoinRun :
>
> | Condition (Δ0–Δ9)          | Permute                                                                           | Window                                                                           | Expand                                                                               |
> |----------------------------|------------------------------------------------------------------------------------|----------------------------------------------------------------------------------|--------------------------------------------------------------------------------------|
> | Standard+SSP+LN            | 0.000, 0.148, 0.162, 0.164, 0.254, 0.310, 0.283, 0.326, 0.373, 0.388              | 0.000, 0.071, 0.060, 0.097, 0.146, 0.200, 0.224, 0.220, 0.277, 0.280             | 0.000, -0.124, -0.137, -0.183, -0.133, -0.098, -0.096, -0.087, -0.112, -0.112       |
> | Standard+SSP+LN+Dropout    | 0.000, 0.123, 0.114, 0.142, 0.191, 0.202, 0.210, 0.174, 0.174, 0.241              | 0.000, 0.169, 0.147, 0.237, 0.263, 0.342, 0.384, 0.415, 0.446, 0.460             | 0.000, -0.120, -0.141, -0.132, -0.129, -0.070, -0.054, -0.061, -0.050, -0.023       |

---

### Official Review · Reviewer_Mb9G · 2025-10-31

**Soundness:** 3
**Presentation:** 3
**Contribution:** 3
**Rating:** 4
**Confidence:** 3

**Summary:**

The paper introduces InterpLayers, an architectural modification aimed at mitigating plasticity loss in continual reinforcement learning (CRL). Instead of relying on algorithmic resets or regularization methods, InterpLayers embed adaptability directly into the network structure. Each layer consists of two paths: a fixed, parameter-free reference pathway that preserves stable representations, and
a learnable projection pathway that adapts through gradient updates. The two pathways are combined via input-dependent interpolation weights (gates) that dynamically control the mix between stability and adaptability.

The authors provide theoretical analysis showing that InterpLayers (1) bound representational drift (churn) and (2) maintain Neural Tangent Kernel (NTK) rank under mild assumptions. Empirically, experiments on ProcGen CoinRun across several distributional shifts (permutation, windowing, and expansion) demonstrate that InterpLayers outperform parameter-matched baselines and perform competitively with Soft Shrink–Perturb + LayerNorm (SSP+LN). The addition of dropout further improves performance under gradual shifts. Overall, the work proposes a simple yet effective architectural alternative to algorithmic interventions for maintaining plasticity in CRL.

**Strengths:**

- The work reframes plasticity preservation as an architectural rather than an optimization problem. By introducing a fixed reference pathway, it provides an elegant, orthogonal solution to methods like resets, normalization, or regularization.
- The paper connects architectural design to measurable stability properties, bounded churn and non-collapsing NTK rank, supported by derivations and intuitive proofs.
- Experiments across multiple ProcGen shifts, with 10 random seeds and diagnostics such as NTK rank and churn, show consistent improvements.
- Results are aligned with the theoretical claims and display robustness.
- InterpLayers introduce negligible computational overhead, require no hyperparameter tuning, and can be easily integrated into existing architectures or combined with other methods.
-This direction, architectural plasticity, is promising and opens up pathways for scalable continual learning systems where task resets or algorithmic perturbations are infeasible.

**Weaknesses:**

- All results are on ProcGen CoinRun, a single-task environment. Other ProcGen games (e.g., CaveFlyer, Heist) or domains such as DMControl or Atari are missing. This makes it unclear whether the gains generalize across input modalities or reward structures.

- The paper mentions “sparse selection or zero-padding” when input and output dimensions differ but does not specify how these projections are chosen. Without an ablation (identity vs. random vs. padded), one cannot tell if the stability comes from architecture or from implicit regularization.
- Methods like Highway Networks, Gated ResNets, FiLM, or LoRA adapters all interpolate between fixed and learned paths. None are compared, which weakens the claim of architectural novelty.
- While dropout improves performance, it’s unclear why? is it due to decorrelation of gates, or mitigation of reference-path bias? An analysis of gate variance or activation entropy under dropout would help.
- Theorem 1 and Theorem 2 bound churn and NTK rank but are non-tight and do not quantify the degree of stability gained. Without empirical correlation (e.g., churn vs performance curve), the theoretical component remains mostly illustrative.
-Plots of gate activations, reference-vs-projection contribution ratios, or NTK spectra evolution could provide intuition about how stability emerges.
- Since one path is fixed, initialization may strongly influence representational geometry. No experiment examines robustness to initialization variance.
- A few formatting and typographical problems (“ae shown”, inconsistent NTK logging frequency—50 vs 100 epochs) slightly reduce polish and precision.

**Questions:**

- Please clarify how P(MLP) and Pr(Conv) are defined when input and output sizes differ. Are they fixed random matrices, identity selections, or sparse maps? How sensitive is performance to these definitions?
- Could you include baselines such as a Gated-ResNet, Highway Network, or Squeeze-and-Excitation module with learnable skip coefficients? These seem architecturally close to InterpLayers and would sharpen the novelty claim.
- Since one branch is frozen from initialization, does the choice of initialization scheme (Xavier, Kaiming, orthogonal) affect stability or gate distribution?
- Have you tested InterpLayers on larger backbones (e.g., deeper CNNs or transformers)? what would be the intuition here?
- Does the fixed path introduce representational bottlenecks as depth increases?
- Do sigmoid gates saturate toward 0 or 1 during long training?
- Beyond FLOPs/parameter count, what is the actual wall-clock increase in training time and memory usage?
- Since LayerNorm affects gradient variance and NTK rank, how does its presence (or absence) interact with InterpLayers’ stability claims?
- If the task sequence is reversed or randomized, does the observed stability persist, or does the architecture implicitly overfit to a specific shift pattern?
- You mention orthogonality to SSP+LN and ReDo. Could you include quantitative results combining InterpLayers with these methods to test additive benefits?
- Have you tried using InterpLayers in supervised continual learning (e.g., permuted MNIST) or offline RL?
- Consider plotting feature-space trajectories or NTK spectra over time to visualize representational stability versus baselines.
- Could you correlate measured churn or NTK-rank changes with actual performance degradation to show causal alignment between theory and outcomes?

**Details Of Ethics Concerns:**

There are no ethics concerns to report.

---

> ### Author Response · Authors · 2025-12-03
>
> The authors are incredibly thankful to the reviewer for your insightful and detailed comments that helped improve our work.
>
> ---
>
> ### Comment 1. (Clarification of the Reference Pathway)
> > *Please clarify how P(MLP) and P(Conv) are defined when input and output sizes differ.*
>
> We thank the reviewer for the insightful comment. In our architecture, neither of P(MLP) nor P(Conv) contains learnable parameters, but are deterministic and structure-preserving mappings.
>
> For linear layers, we have P(MLP) where the reference pathway uses an 'IdentityProject' block. Its function depends on whether the output dimension is smaller, equal, or larger than the input.
>
> If $dim_{out} = dim_{in}$: the operator is the identity.
>
> If $dim_{out} < dim_{in}$: we use a fixed Johnson-Lindenstrauss (JL) projection with orthonormal rows defined as $P \in \mathbb{R}^{out \times in}, P P^\top = I_{out}$, where the seed is fixed per layer so that the map is stable across runs.
>
> If $dim_{out} > dim_{in}$: we use zero-padding. This is inspired by residual architectures that expand dimensions while preserving the identity structure.
>
> For convolutional layers, we have P(Conv) where the reference pathway uses an 'IdentityDownsample' block. If the resolution changes (so stride > 1), we apply average pooling. If channel counts differ, we use channel slicing/padding to match dimensions.
>
> Overall, for both types, the operator used in the reference pathway is deterministic and parameter-free, aiming to preserve spatial structure without introducing any weights. We chose the JL projection for linear layers, as it preserves distances and avoids directional bias, and average pooling for the convolutional layers because it is a standard procedure in residual networks.
>
> We rewrote these definitions in the revised manuscript (lines 194-203) to clarify the definition of P(MLP) and P(Conv).
>
> ---
>
> ### Comment 2. (Gated Networks as a Baseline)
> > *Could you include baselines such as a Gated-ResNet, Highway Network, or Squeeze-and-Excitation module with learnable skip coefficients? These seem architecturally close to InterpLayers and would sharpen the novelty claim.*
>
> Thank you for your comment. According to the reviewer’s suggestion, we add ResNet-like and Highway Networks as baselines. The results show that the majority of tasks and shift types, InterpLayers outperform ResNet-like and Highway Networks in terms of mitigating plasticity loss.
>
> Please check the response to Comment 2 from Reviewer 2fPo and Fig. 2 in the revised manuscript for full results.
>
> ---
>
> ### Comment 3. (Influence of Different Initialization Schemes)
> > *Since one branch is frozen from initialization, does the choice of initialization scheme (Xavier, Kaiming, orthogonal) affect stability or gate distribution?*
>
> We would like to thank the reviewer for this insightful comment.
>
> Our frozen reference pathway does not contain any learnable weights. In the convolutional layers, the reference pathway is an 'IdentityDownsample' block, which performs average pooling and channel padding/slicing and has no parameters. In the linear layers, the reference pathway is an 'IdentityProject' block, which applies a fixed geometric projection following the Johnson-Lindenstrauss lemma. This makes it a structured linear map, not a learned (but frozen) weight matrix. So, there is no initialization of weight parameters using traditional methods like Xavier, Kaiming, or orthogonal initializations.
>
> However, we do believe that the reviewers' intuition does apply to the interpolation weights, where initialization affects their learning stability. In our current framework, interpolation weights are initialized using Kaiming, making their mean centered around 0.5. Here, using a different mean could lead to a bias towards either projection or reference before training starts. Initializing with a mean of 0.5, we try to avoid introducing bias into the gates and make their values purely driven by the task.
>
> In future studies, we aim to explore if introducing such bias or changing the initial scale of the gate logits has an impact on the performance and plasticity retention of InterpLayers. We do believe that finding a robust choice here is important in continual learning settings.
>
>
> ---
>
> ### Comment 4. (Larger Backbone Networks)
> > *Have you tested InterpLayers on larger backbones (e.g., deeper CNNs or transformers)? What would be the intuition here?*
>
> Thank you for your insightful comment. We have not tested InterpLayers on larger backbones due to computational constraints on our server. Theoretically, InterpLayers are scalable to larger backbones without any changes in their formulation. However, additional empirical evaluation is needed to verify the effective learning of the interpolation mechanism for deeper layers.

---

> > ### Author Response · Authors · 2025-12-03
> >
> > ### Comment 5. (Representational Bottlenecks)
> > > *Does the fixed path introduce representational bottlenecks as depth increases?*
> >
> > This is an interesting question. As the reference pathway is fixed, as depth increases, the information should pass through many fixed transformations. In our case, as discussed in our answer to Comment 3, these transformations aim to preserve scale and geometric structure. Complementary to the reference pathway, our InterpLayer architecture has a learnable projection pathway and interpolation weights, which are not restricted or biased toward either pathway. So, theoretically, InterpLayers have properties that do not incorporate representation bottlenecks. Here, the intuition is that $z$, the interpolation weight, is adjusted dynamically to increase the effect of the projection if necessary to maintain plasticity.
> >
> > We believe, though, that the reviewer’s point is very important. For example, we observe that dropout enhances the InterpLayer performance, as it keeps a level of variance between reference and projection that avoids representational bottlenecks. Additionally, a direction for further evaluation is whether, when applying InterpLayers for deep networks, direct skip connections from input to deep layers change the interpolation dynamics across the entire network. Although our empirical results for the different shifts did not indicate any kind of performance degradation, we think that a better understanding of how the reference pathway acts with deeper layers will improve the robustness of InterpLayers such that complementary techniques like dropout are not necessary.
> >
> > ---
> >
> > ### Comment 6. (Characteristics of the Interpolation Mechanism)
> > > *Do sigmoid gates saturate toward 0 or 1 during long training?*
> >
> > According to the reviewer’s suggestion, we expand our discussion regarding the characteristics of the magnitudes of the output of sigmoid gates in the Discussion section of the revised manuscript (lines 421-426). We see, as shown in Fig. 4, that the gates saturate with values around 0.2, so towards 0, slightly favouring the reference pathway.
> >
> > In addition to the mean gate value for early and late tasks, we now include the full mean trajectory and the variability of these values across training in Appendix H. In the trajectory, we show that the means do stabilize around 0.2 and are not decreasing sharply to 0. To obtain the variance, we calculate the normalized diversity ratio of each gate distribution:
> >
> > $NGDR = \frac{Var[z_t]}{\mu_t (1 - \mu_t)}$,
> >
> > where $z_t$ represents the interpolation weight distribution in a given layer at epoch $t$, and $\mu_t$ is its mean. The denominator is the variance of a Bernoulli distribution with mean $\mu_t$, so that this normalized gate diversity ratio (NGDR) metric works as a scale-free measure of how diverse the interpolation weights are relative to the maximal possible variance.
> >
> > Across all shifts and tasks, we see that although the gates saturate toward values around 0.2 (as mentioned above), their NGDR remains stable (between 0.3 and 0.5) over training. This suggests that the interpolation weights do not collapse to a single value but instead remain stable or even increase their variance. This empirically supports the mechanism behind Theorem 2. Interpolation weight distributions maintain non-degenerate variance, which consequently preserves gradient diversity and a stable effective NTK rank.
> >
> > ---
> >
> > ### Comment 7. (Increase in Wall-Clock Time)
> > > *Beyond FLOPs/parameter count, what is the actual wall-clock increase in training time and memory usage?*
> >
> > Thank you for your suggestion. Compared against other baselines, InterpLayers do not show an increase in memory usage but a 10% increase in wall-clock training time. However, it is hard to determine how much of that increase is due to architectural changes and how much is due to the additional logging of gating statistics used to perform the analysis in Section 4.4. In summary, the wall-clock time from InterpLayers seems comparable to that of other top-performing baselines such as SSP+LN and Highway Networks. The detailed results are shown in the next table that is added to Appendix C in the revised manuscript.
> >
> > ### Training Cost Summary
> >
> > | Condition                | # Runs | Avg Time/Run (hrs)  | Avg Final Memory (MB) |
> > |--------------------------|--------|---------------------|------------------------|
> > | Highway                  | 120    | 23.65               | 8265.8                 |
> > | ResNet-like              | 120    | 21.55               | 8318.2                 |
> > | Standard+SSP+LN          | 120    | 23.71               | 8309.6                 |
> > | Standard+Dropout         | 120    | 21.68               | 8305.2                 |
> > | InterpLayers (Ours)      | 120    | 25.78               | 8311.8                 |
> >
> > ---

---

> > > ### Author Response · Authors · 2025-12-03
> > >
> > > ### Comment 8. (Influence of LayerNorm)
> > > > *Since LayerNorm affects gradient variance and NTK rank, how does its presence (or absence) interact with InterpLayers’ stability claims?*
> > >
> > > Thank you for your question. We ran an ablation study combining InterpLayers with LayerNorm. The results show that LayerNorm improves the performance of InterpLayers, suggesting that this combination improves gradient variance and NTK rank. We add this ablation study to Appendix M in the revised manuscript.
> > >
> > > ### Comment 9. (Robustness of InterpLayers)
> > > > *If the task sequence is reversed or randomized, does the observed stability persist?*
> > >
> > > Thank you for your comment. We conducted an ablation study where we randomized the task order. For example, the original task order ‘1-2-3-...-10’ was permuted to a new task order ‘1-5-3-...-2’. The results obtained showed a similar level of performance retention and raw reward magnitudes for both task orders. These results suggest that the proposed architecture is robust for different task orders, a crucial characteristic of continual learning algorithms.
> > >
> > > ---
> > >
> > > ### Comment 10. (Orthogonality of InterpLayers)
> > > > *Include quantitative results combining InterpLayers with SSP+LN and ReDo.*
> > >
> > > Thank you for your suggestion. Additionally to the comparison with InterpLayer-LN we also ran an ablation study combining InterLayers and SSP+LN. The results show that the performance of InterpLayer-LN and InterpLayer-SSP+LN were very similar. These results improve the robustness of our claim that InterpLayers are complementary to existing methods. We add this ablation study to Appendix L in the revised manuscript.
> > >
> > > ### Comment 11. (InterpLayers in Supervised CL and Offline RL)
> > > > *Have you tried using InterpLayers in supervised continual learning (e.g., permuted MNIST) or offline RL?*
> > >
> > > Thank you so much for your suggestion. We have not attempted to use InterpLayers in supervised continual learning or offline RL. We believe that InterpLayers would also be well-suited, given the similarity in the properties of the permuted MNIST task and the ProcGen tasks. This is an interesting direction for future exploration.
> > >
> > > ---
> > >
> > > ### Comment 12. (Representational Stability)
> > > > *Plot NTK spectra or provide explanation.*
> > >
> > > Thank you for the interesting suggestion. To plot the NTK spectra, we need to store all the gradients during training, which was not feasible at the moment due to our server constraints. We are excited to perform this analysis as future work.
> > >
> > > ---
> > >
> > > ### Comment 13. (Correlation between Theoretical Metrics and Performance)
> > > > *Could you correlate measured churn or NTK-rank changes with actual performance degradation?*
> > >
> > > According to the reviewer’s suggestion, we have analyzed the correlation between the measured churn and NTK-rank changes with actual performance during training. Although these properties are crucial for maintaining plasticity, we did not find any correlation when analyzing these correlations at an epoch level for the entire training. However, as shown in the Section 4.3 of the manuscript, we did observe that methods that display a lower churn tend to have better performance retention overall.
> > >
> > > ---
> > >
> > > ### Comment 14. (Minor Comments)
> > > > *A few formatting and typographical problems.*
> > >
> > > Thank you for your detailed review. We have addressed these issues in the revised manuscript.
> > >
> > > ---

---

### Official Review · Reviewer_mpmB · 2025-11-01

**Soundness:** 2
**Presentation:** 2
**Contribution:** 2
**Rating:** 2
**Confidence:** 4

**Summary:**

This work studies Continual RL, especially focusing on the issue of plasticity loss. This paper proposes InterpLayers, a drop-in architectural solution to plasticity loss, to improve the learning performance of CRL. The key idea of InterpLayers is to build a fixed reference pathway and a learnable projection for each layer, which are further interpolated by a state-dependent learnable parameter. The experiments are conducted in CoinRun with three CRL settings. The results include performance comparison and the analysis of churn, NTK, and the interpolation weights learned.

**Strengths:**

- The proposed method is clearly presented.
- The experiments provide multi-faceted results, including performance comparison and the analysis of churn, NTK, and the interpolation weights learned.
- The related works are satisfactorily discussed. Some more related works should be included, like [1,2] for plasticity loss study, and [3,4] for network architecture study.

---
### Reference

[1] Directions of Curvature as an Explanation for Loss of Plasticity. https://arxiv.org/abs/2312.00246

[2] Deep Reinforcement Learning with Plasticity Injection. https://arxiv.org/abs/2305.15555

[3] Hyperspherical Normalization for Scalable Deep Reinforcement Learning. https://arxiv.org/abs/2502.15280

[4] Bigger, Regularized, Optimistic: scaling for compute and sample-efficient continuous control. https://arxiv.org/abs/2405.16158

**Weaknesses:**

- I feel that the proposed method is not well motivated in this draft. I did not get the point of the proposed method until I finished reading Section 3.2. I think using some illustrative examples will help convey the motivation and key idea.
- The InterpLayer has a similar architecture with Residual layer. It seems that the major difference is the frozen reference pathway and a learnable interpolation weight $z$. The significance of the difference should be elaborated and empirically examined.
- The experiments should be strengthened:
    - My major concern is that only one environment is used (i.e., CoinRun), which is insufficient to provide a convincing evaluation.
    - The readability of Figure 2 and Figure 3 should be improved, e.g., fontsize, color (red and pink/purple are hard to tell).
    - InterpLayer by itself does not work well according to the green and yellow lines in Figure 2.
    - The authors mentioned that the proposed method is a complementary solution to existing methods. However, InterpLayer is not combined with existing methods to examine its orthogonal benefit.

### Minors

- “(Abbas et al., 2023) investigated weight clipping to provide an upper bound to parameter growth”. Incorrect reference.

**Questions:**

1. Since the InterpLayer has a similar architecture with Residual layer (the major difference seems to be the frozen reference pathway and a learnable parameter $z$), I recommend that the authors include residual layer as a baseline in the experiments.
2. It seems that InterpLayer by itself does not work well as shown in Figure 2. Does it mean dropout is necessary to be used together with InterpLayer?
3. Could the authors provide the results for InterpLayer + SSP + LN, as the authors mention that the proposed method is a complementary solution to existing methods?
4. In my opinion, only using CoinRun is not enough to obtain convincing experimental results. Could the author provide (at least) 3 more Procgen environments?

---

> ### Author Response · Authors · 2025-12-03
>
> The authors are incredibly thankful to the reviewer for your insightful and detailed comments that helped improve our work.
>
> ---
>
> ### Comment 1. (Residual Layer as a Baseline)
> > *Since the InterpLayer has a similar architecture with Residual layer, I recommend that the authors include residual layer as a baseline in the experiments.*
>
> Thank you for your comment. According to the reviewer’s suggestion, we add a ResNet-like architecture as an additional baseline. The results show that across the majority of tasks and shift types, InterpLayers outperform ResNet-like structures in terms of performance retention.
>
> Please check the response to Comment 2 from Reviewer 2fPo and Fig. 2 in the revised manuscript for full results.
>
> ---
>
> ### Comment 2. (Dropout)
> > *It seems that InterpLayer by itself does not work well as shown in Figure 2. Does it mean dropout is necessary to be used together with InterpLayer?*
>
> Thank you for your insightful comment. Applying dropout to the projection pathway of InterpLayers guarantees some variance between reference and projection, which is crucial for learning effective interpolation weights. We expand on this intuition as follows.
>
> Different from the standard application in neural networks, where dropout masks neurons in all layers, in our method, we apply dropout only to the projection pathway. Our intuition is that applying dropout only to the projection increases the activation variance of the projection pathway, affecting the representational gap $D$ defined in Theorem 1.
>
> In Theorem 1, we show that InterpLayers provide an upper bound to their output variability by:
>
> $\left\lVert \Delta h(x) \right\rVert_{2}
> \le
> \lVert z(x) \rVert_{\infty} L_{p} \lVert \Delta \theta_{p} \rVert_{2}
> +
> L_{z} \lVert \Delta \theta_{z} \rVert_{2} D(x),
> \quad
> D(x) = \left\lVert h_{\mathrm{proj}}(x) - h_{\mathrm{ref}}(x) \right\rVert_{2}.$
>
> In this equation, the term $D$ determines how much the changes in the interpolation weights affect the output. In this case, if the projection and reference are equal, the interpolation weights do not have an impact on the output. However, if $D$ increases, the impact of the interpolation pathway also increases. Dropout strictly increases the activation variance in the projection pathway by injecting noise. So the instantaneous gap is as follows
>
> $D_t(x) = \bigl\|| \tilde{h}_{\mathrm{proj},t}(x) - h _ {\mathrm{ref}} (x)\bigr\||.$
>
> This way, dropout guarantees that $D_t(x)$ always has some variance, stabilizing the gradient of the interpolation pathway z(x) in Eq. 7.
>
> We added this additional discussion to Appendix H of the revised manuscript.
>
> ---
>
> ### Comment 3. (Orthogonality of InterpLayers)
> > *Could the authors provide the results for InterpLayer + SSP + LN, as the authors mention that the proposed method is a complementary solution to existing methods?*
>
> Thank you for your suggestion. We ran an ablation study to compare the standalone InterpLayer implementation with InterpLayer-LN and InterpLayer-SSP+LN. The results show that combining InterpLayers with LN improves initial convergence speed when compared to standalone InterpLayers. Interestingly, the performance of InterpLayer-LN and InterpLayer-SSP+LN was very similar. These results improve the robustness of our claim that InterpLayers are complementary to existing methods. We add this ablation study to Appendix M in the revised manuscript.
>
> ---
>
> ### Comment 4. (Limited Experimental Evaluation)
> > *In my opinion, only using CoinRun is not enough to obtain convincing experimental results. Could the author provide (at least) 3 more Procgen environments?*
>
> Thank you for your comment. According to the reviewer’s suggestion, we performed the experiments for three more ProcGen environments, Jumper, Fruitbot, and Heist. The results show that for the majority of tasks and shift types, InterpLayers outperform ResNet-like architectures and Highway Networks in terms of mitigating plasticity loss and achieve comparable performance when compared to SSP+LN.
>
> Please check the response to Comment 2 from Reviewer 2fPo and the figures in the revised manuscript for full results.
>
> ---
>
> ### Comment 5. (Improve Figures)
> > *The readability of Figure 2 and Figure 3 should be improved.*
>
> Thank you for your comment. We have revised the figures to improve readability in the revised manuscript.
>
> ---
>
> ### Comment 6. (Related Works)
> > *Some more related works should be included.*
>
> Thank you for bringing these interesting works to our attention. We have added them to the Related Works section of our manuscript.
>
> ---
>
> ### Comment 7. (Incorrect Reference)
> > *Incorrect reference: (Abbas et al., 2023)*
>
> Thank you for your detailed review. We have corrected this reference in the revised manuscript.
>
> ---

---

### Official Review · Reviewer_2fPo · 2025-11-04

**Soundness:** 2
**Presentation:** 2
**Contribution:** 2
**Rating:** 2
**Confidence:** 3

**Summary:**

This paper proposes InterpLayers, an architectural solution for plasticity loss in continual reinforcement learning. Each InterpLayer combines a fixed reference pathway with a learnable projection pathway using input-dependent interpolation weights. The authors provide theoretical analysis showing bounded churn and NTK rank preservation, and evaluate on three ProcGen distribution shifts (permute, window, expand), demonstrating that InterpLayers with dropout achieve comparable or better performance than baselines including SSP+LN.

**Strengths:**

- The architectural approach is clean and task-agnostic, requiring no task boundaries, reset schedules, or additional hyperparameters beyond standard training, making it easy to integrate into existing systems.
- Theoretical analysis provides formal guarantees for bounded representational drift (Theorem 1) and NTK rank preservation under variance assumptions (Theorem 2), connecting architectural design to plasticity metrics.
- Empirical evaluation includes comprehensive ablations examining visual uncertainty, entropy bonuses, annealed sampling, and dropout interactions, with analysis of theoretical metrics (churn, NTK rank) validating the theoretical predictions.

**Weaknesses:**

- Experimental evaluation is severely limited in scope, testing only on ProcGen coinrun with three distribution shift types. The paper lacks evaluation on standard CRL benchmarks (Atari suite, other ProcGen games, MuJoCo control) that would demonstrate generalizability and enable fair comparison with the broader continual learning literature.
- Performance improvements are modest and inconsistent across conditions. SSP+LN sometimes outperforms InterpLayers (e.g., permute task), and the best results require combining InterpLayers with dropout, suggesting the core mechanism alone is insufficient. The paper provides no clear guidance on when conv-only versus fullinterp should be used.

**Questions:**

- Can you provide results on standard CRL benchmarks (Atari, other ProcGen environments, MuJoCo continuous control) to demonstrate the method's generalizability and enable comparison with the broader literature?
- The best results require combining InterpLayers with dropout, which suggests the core mechanism may be insufficient. Can you provide deeper analysis of why dropout is necessary and whether InterpLayers provide benefits beyond standard networks with dropout?

---

> ### Author Response · Authors · 2025-12-03
>
> The authors are incredibly thankful to the reviewer for your insightful and detailed comments that helped improve our work.
>
> ---
>
> ### Comment 1. (Limited Experimental Evaluation)
>
> > *Can you provide results on standard CRL benchmarks (Atari, other ProcGen environments, MuJoCo continuous control) to demonstrate the method's generalizability and enable comparison with the broader literature?*
>
> Thank you for your comment. According to the reviewer’s suggestion, we performed the experiments for three more ProcGen environments, Heist, Fruitbot, and Jumper. The results show that for the majority of tasks and shift types, InterpLayers outperform ResNet-like architectures and Highway Networks in terms of mitigating plasticity loss and achieve comparable performance when compared to SSP+LN.
>
> Next, we add tables showing the full results on performance retention (higher is better) for the 4 ProcGen environments evaluated in our work. The figures, main text, and appendix are updated in the revised manuscript to reflect these results.
>
> ---
>
> ### CoinRun:
>
> | Condition (Δ0–Δ9)                  | Permute                                                                        | Window                                                                         | Expand                                                                         |
> | --------------------------- | ----------------------------------------------------------------------------- | ----------------------------------------------------------------------------- | ----------------------------------------------------------------------------- |
> | Highway                     | 0.000, -0.127, -0.138, -0.242, -0.195, -0.336, -0.238, -0.412, -0.400, -0.347 | 0.000, -0.022, -0.061, -0.727, -0.910, -0.451, -0.305, -0.404, -0.252, -0.413 | 0.000, -0.238, -0.472, -1.027, -2.096, -1.006, -2.798, -1.682, -1.894, -1.726 |
> | ResNet-like                 | 0.000, -0.233, -0.532, -0.467, -0.437, -0.486, -0.453, -0.233, -0.376, -0.479 | 0.000, -0.108, -0.339, -0.263, -0.385, -0.802, -0.543, -1.010, -0.791, -1.000 | 0.000, -0.342, -0.783, -1.064, -1.570, -2.600, -2.817, -2.098, -1.700, -2.159 |
> | InterpLayers (Ours) | 0.000, -0.023, -0.064, -0.019, -0.050, 0.101, 0.132, 0.207, 0.144, 0.107      | 0.000, 0.029, 0.093, 0.248, 0.324, 0.430, 0.477, 0.488, 0.563, 0.552          | 0.000, -0.220, -0.237, -0.178, -0.177, -0.083, -0.032, 0.003, 0.049, 0.073    |
> | Standard+Dropout         | 0.000, 0.252, 0.263, 0.280, 0.271, 0.192, 0.227, 0.282, 0.233, 0.226          | 0.000, 0.190, 0.224, 0.223, 0.228, 0.279, 0.291, 0.308, 0.335, 0.345          | 0.000, -0.062, -0.108, -0.079, -0.116, -0.129, -0.120, -0.092, -0.071, -0.085 |
> | Standard+SSP+LN             | 0.000, 0.148, 0.162, 0.164, 0.254, 0.310, 0.283, 0.326, 0.373, 0.388          | 0.000, 0.071, 0.060, 0.097, 0.146, 0.200, 0.224, 0.220, 0.277, 0.280          | 0.000, -0.124, -0.137, -0.183, -0.133, -0.098, -0.096, -0.087, -0.112, -0.112 |
>
> ---
>
>
>
> ### Heist:
>
> | Condition (Δ0–Δ9)            | Permute                                                                      | Window                                                                         | Expand                                                                         |
> | --------------------------- | ----------------------------------------------------------------------------- | ----------------------------------------------------------------------------- | ----------------------------------------------------------------------------- |
> | Highway                     | 0.000, 0.034, 0.072, -0.224, -0.374, -0.608, -0.663, -0.835, -0.913, -1.058   | 0.000, -0.213, -0.326, -0.296, -0.025, -0.314, -0.385, -0.115, 0.129, -0.145  | 0.000, -0.776, -1.167, -1.220, -1.784, -1.926, -1.695, -1.786, -1.754, -1.590 |
> | ResNet-like                 | 0.000, 0.309, 0.338, 0.254, -0.766, -0.891, -1.809, -2.593, -2.153, -2.252    | 0.000, -0.182, -0.286, -0.237, 0.108, -0.180, -0.893, -2.609, -2.214, -2.699  | 0.000, -0.800, -1.393, -1.606, -1.973, -2.213, -2.134, -2.207, -3.169, -3.207 |
> | InterpLayers (Ours) | 0.000, 0.300, 0.442, 0.542, 0.309, 0.283, 0.172, 0.149, -0.047, -0.145        | 0.000, 0.034, 0.069, 0.140, 0.442, 0.427, 0.620, 0.924, 0.979, 0.932          | 0.000, -0.785, -0.932, -0.743, -0.956, -0.660, -0.368, -0.199, 0.058, 0.210   |
> | Standard+Dropout        | 0.000, -0.180, -0.415, -0.740, -1.052, -1.185, -1.401, -1.563, -1.551, -1.734 | 0.000, -0.288, -0.405, -0.773, -0.805, -1.166, -1.328, -1.125, -1.183, -1.341 | 0.000, -1.160, -1.799, -2.135, -2.845, -3.089, -3.270, -3.270, -3.306, -3.199 |
> | Standard+SSP+LN            | 0.000, 0.732, 0.959, 1.173, 1.187, 1.360, 1.424, 1.389, 1.485, 1.517          | 0.000, 0.782, 0.949, 1.168, 1.422, 1.482, 1.583, 1.613, 1.733, 1.629          | 0.000, 0.217, 0.150, 0.130, -0.400, -0.470, -0.394, -0.420, -0.512, -0.438    |
>
> ---

---

> > ### Author Response · Authors · 2025-12-03
> >
> > ### FruitBot:
> >
> > | Condition (Δ0–Δ9)                   | Permute                                                                        | Window                                                                         | Expand                                                                         |
> > | --------------------------- | ----------------------------------------------------------------------------- | ----------------------------------------------------------------------------- | ----------------------------------------------------------------------------- |
> > | Highway                     | 0.000, 0.165, 0.162, 0.224, 0.203, 0.307, 0.311, 0.220, 0.178, -0.042         | 0.000, 2.721, 3.506, 2.639, 0.988, 0.865, 0.036, -0.469, -1.608, -1.404       | 0.000, 1.956, 2.674, 2.497, 1.733, 0.952, 0.778, -0.524, -0.544, -1.140       |
> > | ResNet-like                 | 0.000, -0.631, -1.199, -1.255, -2.019, -2.066, -2.948, -2.537, -2.730, -2.896 | 0.000, -1.626, -1.944, -2.469, -2.891, -3.314, -2.698, -3.723, -3.863, -3.495 | 0.000, -1.845, -2.725, -2.755, -3.095, -2.978, -2.886, -3.228, -3.909, -3.992 |
> > | InterpLayers (Ours) | 0.000, 0.366, 0.470, 0.493, 0.580, 0.505, 0.582, 0.531, 0.536, 0.466          | 0.000, 2.491, 3.910, 4.290, 4.356, 4.357, 4.756, 4.677, 4.604, 4.533          | 0.000, 1.677, 2.818, 3.287, 3.323, 3.512, 3.572, 3.562, 3.619, 3.694          |
> > | Standard+Dropout       | 0.000, 0.183, 0.197, 0.122, 0.167, 0.180, 0.148, 0.085, 0.066, 0.036          | 0.000, 2.259, 3.802, 4.348, 4.256, 4.260, 4.153, 4.192, 4.142, 4.184          | 0.000, 1.647, 2.583, 2.879, 2.908, 2.779, 2.787, 2.771, 2.783, 2.961          |
> > | Standard+SSP+LN            | 0.000, 0.211, 0.250, 0.265, 0.240, 0.286, 0.335, 0.294, 0.394, 0.319          | 0.000, 2.948, 4.803, 5.258, 5.294, 5.452, 5.447, 5.394, 5.390, 5.257          | 0.000, 2.119, 3.513, 4.017, 4.120, 4.162, 4.119, 3.974, 3.883, 3.914          |
> >
> > ---
> >
> > ### Jumper :
> >
> > | Condition (Δ0–Δ9)                   | Permute                                                                         | Window                                                                   | Expand                                                                         |
> > | --------------------------- | ----------------------------------------------------------------------------- | ----------------------------------------------------------------------- | ----------------------------------------------------------------------- |
> > | Highway                     | 0.000, 0.035, 0.020, 0.033, 0.019, -0.048, -0.040, -0.064, -0.121, -0.074     | 0.000, 0.209, -0.105, -0.120, 0.008, 0.287, 0.391, 0.343, 0.178, 0.431  | 0.000, 0.074, -0.104, -0.222, -0.294, -0.204, -0.120, -0.229, -0.215, -0.140  |
> > | ResNet-like                 | 0.000, -0.561, -0.641, -0.689, -0.618, -0.655, -0.591, -0.629, -0.561, -0.499 | 0.000, 0.211, -0.042, -0.112, -0.099, 0.109, 0.372, 0.340, 0.054, 0.384 | 0.000, -0.115, -0.308, -0.444, -0.524, -0.459, -0.361, -0.401, -0.407, -0.312 |
> > | InterpLayers (Ours) | 0.000, 0.387, 0.455, 0.489, 0.445, 0.434, 0.436, 0.444, 0.429, 0.400          | 0.000, 0.545, 0.198, 0.117, 0.268, 0.396, 0.608, 0.614, 0.433, 0.678    | 0.000, 0.034, -0.169, -0.280, -0.395, -0.482, -0.405, -0.458, -0.563, -0.421  |
> > | Standard+Dropout       | 0.000, 0.268, 0.257, 0.245, 0.221, 0.256, 0.209, 0.254, 0.250, 0.281          | 0.000, 0.447, 0.112, 0.025, 0.133, 0.445, 0.611, 0.520, 0.315, 0.595    | 0.000, 0.279, 0.114, -0.023, -0.181, -0.216, -0.063, -0.234, -0.234, -0.177   |
> > | Standard+SSP+LN            | 0.000, 0.041, -0.067, -0.088, 0.072, 0.332, 0.485, 0.472, 0.277, 0.625        | 0.000, 0.237, -0.067, -0.088, 0.072, 0.332, 0.485, 0.472, 0.277, 0.625  | 0.000, 0.065, -0.122, -0.198, -0.322, -0.230, -0.086, -0.210, -0.195, -0.130  |
> >
> >
> > ---

---

> ### Author Response · Authors · 2025-12-03
>
> ### Comment 2. (Dropout)
> >*Can you provide deeper analysis of why dropout is necessary and whether InterpLayers provide benefits beyond standard networks with dropout?*
>
> Thank you for your insightful comment. Different from the standard application in neural networks, where dropout masks neurons in all layers, in our method, we apply dropout only to the projection pathway. Our intuition is that applying dropout only to the projection increases the activation variance of the projection pathway, affecting the representational gap $D$ defined in Theorem 1.
>
> In Theorem 1, we show that InterpLayers provide an upper bound to their output variability by:
>
> $\left\lVert \Delta h(x) \right\rVert_{2}
> \le
> \lVert z(x) \rVert_{\infty} L_{p} \lVert \Delta \theta_{p} \rVert_{2}
> +
> L_{z} \lVert \Delta \theta_{z} \rVert_{2} D(x),
> \quad
> D(x) = \left\lVert h_{\mathrm{proj}}(x) - h_{\mathrm{ref}}(x) \right\rVert_{2}.$
>
>
>
> In this equation, the term $D$ determines how much the changes in the interpolation weights affect the output. In this case, if the projection and reference are equal, the interpolation weights do not have an impact on the output.
>
> However, if $D$ increases, the impact of the interpolation pathway also increases. Dropout strictly increases the activation variance in the projection pathway by injecting noise. So the instantaneous gap is as follows:
>
> $D_t(x) = \bigl\|| \tilde{h}_{\mathrm{proj},t}(x) - h _ {\mathrm{ref}} (x)\bigr\||.$
>
>
>
>
>
> This way, dropout guarantees that $D_t(x)$ always has some variance, stabilizing the gradient of the interpolation pathway $z(x)$ in Eq. 7.
>
> **InterpLayers with Dropout vs Standard Networks with Dropout.** In Fig. 2 of the manuscript, we compare InterpLayers with standard networks with dropout. We show that applying dropout also improves the plasticity of standard networks significantly. However, it does not fully prevent plasticity loss across all conditions, and it does not outperform InterpLayers. We hypothesize that this difference in performance happens because standard layers do not benefit from the additional activation variance as InterpLayers, which contain both a reference and a projection pathway. Instead, for standard networks, dropout decreases model-level variance, which slows down the performance collapse but does not prevent it, as the empirical results suggest.
>
> We added this additional discussion to Appendix H on the revised manuscript.
>
> ---
>
> ### Comment 3. (Architecture Choice)
> >*The paper provides no clear guidance on when conv-only versus fullinterp should be used.*
>
> Thank you for your insightful comment. Theoretically, InterpLayers can be applied to any layer in the network. In the architecture used by the policy in our environment, convolutional layers act as a feature encoder, and a final linear layer acts to combine these features. We use two InterpLayer variants: (i) conv-only (our default variant): where InterpLayers are applied only to the feature encoder; and (ii) fullinterp: where InterpLayers are applied to all layers. Because of this difference, the conv-only variant reduces the parameter count by 524,544 in our current architecture. Our empirical results show that applying InterpLayers only to the feature encoder (conv-only) performs equally or outperforms the fullinterp variant (Appendix K).
>
> According to the reviewer’s comment, we added to the manuscript (Appendix I) a guide on how to choose which InterpLayer variant to use. In summary, for architectures with a clear feature encoder and in applications in which computational efficiency is critical, applying InterpLayers only in the feature encoder part (like in conv-only) might be desired. In scenarios where there are no computational restrictions and the training becomes unstable after task changes, using InterpLayers across all layers of the network is desirable.
>
> ---

---

### Author Response · Authors · 2025-12-03
**General Response**

We would like to thank the reviewers for their insightful feedback that helped improve our work. We would also like to thank the ACs for their effort during this rebuttal period. We really appreciate the extra effort being made to keep the integrity of the peer review process. Thank you.

We are grateful for the strengths of our work highlighted by the reviewers:

* The architecture provides a clean and elegant solution to plasticity loss (2fPo, Mb9G, N1pd)
* The experiments offer multi-faceted results aligned with theoretical claims (mpmB, Mb9G, N1pd)
* The paper connects architectural design to measurable stability properties (Mb9G)

In response to the reviewer’s comments, we conducted experiments in additional environments and with new baselines that helped us better address the concerns regarding limited experimental evaluation and improve the robustness of the theoretical aspects of architectural plasticity in continual learning. Our response to the reviews focuses on these three main points that are also updated in the revised manuscript:

* Expand our experiments with 3 new ProcGen environments, additional gated architecture baselines, and ablation studies
* Clarification on the need of Dropout in the proposed InterpLayers architecture
* Clarification of the theoretical aspects of InterpLayers and overall improvement of the paper content

We have addressed the reviewer’s concerns to the best of our abilities and highlighted the changes in the revised manuscript to clarify the parts changed based on the feedback.

Sincerely,
Authors

---

### Meta-Review · Area_Chair_whu2 · 2026-01-04

**Summary:**

I recommend a reject. This paper seems not ready for this venue. All reviewers pointed out that the evaluation is limited to one environment, which was later expanded to three more, which is still modest. Moreover, overall performance is modest. I strongly recommend the authors to expand the evaluation to other task suites and look for ways to improve performance in large-scale experiments.

**Reviewer Concerns:**

Main concerns:

All reviewers:
- Experimental evaluation is severely limited in scope: Evaluation only on ProcGen CoinRun and missing broader CRL benchmarks.
- - Authors added three more ProcGen environments

Reviewer 2fPo:
- Performance improvements are modest and inconsistent across conditions
- - Authors acknowledged that baseline SSP+LN leads in raw reward overall.

**Reviewer Scores:**

Given that the performance is mixed and evaluation is limited, it is unlikely that any of the scores would have significantly increased.

---

### Decision · Program_Chairs · 2026-01-26

Reject